# General Markov Model for Solving Patrolling Games

**Andrzej Nagórko**[1,2]    **Marcin Waniek**[1,2]    **Małgorzata Róg**[1]

**Michał Godziszewski**[1,2]    **Barbara Rosiak**[1]    **Tomasz P. Michalak**[1,2]

[1]IDEAS NCBR, Warsaw, Poland

[2]Faculty of Mathematics, Informatics and Mechanics, University of Warsaw, Warsaw, Poland

## Abstract

Safeguarding critical infrastructure has emerged as a global challenge. Effective mobile security forces are essential to address complex security concerns. A key challenge involves designing optimal patrolling strategies for mobile units. Two bodies of research dealt with this: stochastic patrolling and partially observable stochastic games. Alas, the first approach makes too-far-reaching simplifying assumption and the second one is computationally challenging. The model proposed in this paper is inspired by partially observable stochastic games, and it enables comprehensive modeling of attacker-defender interactions while remaining computationally friendly. With our robust SHIELD algorithm, we are able to find a defense strategy where the probability of capturing the attacker can be nearly doubled compared to the state of the art.

## 1  INTRODUCTION

In an era of growing connectivity and technological interdependence, protecting critical infrastructure has become a worldwide challenge. Unfortunately, recently rising geopolitical instabilities have made maintaining desirable security level significantly more difficult [Tamilselvan et al., 2024]. While in the last decades threats consisted of crime, industrial espionage, or terrorism, nations now have to safeguard critical infrastructure from state-sponsored (hybrid-warfare) attacks, exemplified by incidents in the Baltic Sea [Bueger and Liebetrau, 2023]. This issue is exacerbated by advancing technology that allows for more sophisticated attacks in remote locations. This increases the sheer size of the areas that have to be protected. For instance, a single offshore wind farm in the Baltic Sea typically covers an area of about $100 \, \text{km}^2$, with dozens of wind turbines, offshore transformer stations, and hundreds of kilometers of underwater cables.

Last but not least, our understanding of what should constitute critical infrastructure has evolved and it is now much broader [Pursiainen and Kytömaa, 2023]. As a result, any limited security resources has to be spread even thinner.

An effective security force needs to be mobile, enabling it not only to detect an attack but also to promptly summon an appropriate response. In this context, one of the key issues is to design optimal patrolling strategies for mobile units. Unfortunately, under realistic assumptions this becomes a challenging game-theoretic problem. Deterministic routes are predictable; thus, they are likely to be exploited by an attacker. Given this, the literature focused on the so-called stochastic patrolling, where the defender(s) randomize their behaviour [Basilico, 2022]. The time in the model is discretized into turns during which both players, the defender and the attacker, take actions. The attacker observes the moves of the defender and can attack any target at any turn, but the penetration takes a predefined number of turns. If detected within this time, the attack fails. A recent work in this vein, John et al. [2023], studies the problem of patrolling San Francisco intersections by a police unit. The authors assume that the defender strategy is a standard Markov decision process and that this strategy is known by the attacker including defender's current position. Unfortunately, the stochastic patrolling literature introduces a plethora of simplifying assumptions that make its results difficult to apply in a realistic setting. For instance, while the capabilities of the defender and the attacker are in reality asymmetric, in the stochastic patrolling literature they usually make their decision based on the same (or very similar) information.

This limitation can be to some extent addressed by employing partially observable stochastic games (POSGs) [Horák et al., 2023]. Specifically, POSGs enable explicit modelling of the information asymmetry between the attacker and the defender, allowing one party to observe a smaller subset of the environment. Within this framework, both the attacker and the defender can perform multiple actions, modifying the state of the environment. Moreover, both players make their decisions based on the entire history of the confronta-

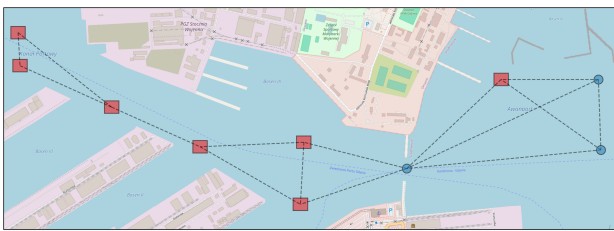

Figure 1: Stylized scenario of defending the port of Gdynia.

tion, with a discount factor giving greater weight to current events. However, all this additional expression power comes at a cost of a greater computational challenge, making POSGs difficult to apply in practice.

In this work we present a model that combines methods from stochastic patrolling [Duan et al., 2019, 2021, John et al., 2023] based on a Markov chain control with game-theoretic methods based both on POSGs [Horák et al., 2023] and on Stackelberg games that were successfully applied in many real-world scenarios [Pita et al., 2009, Shieh et al., 2012]. For the new model, we construct an effective algorithm to compute an optimal strategy in a scenario when the security incidents are scarce and there is a little interaction between the defender and the attacker, but a successful attack is devastating. In particular, we consider an infinite event horizon with no discount factor and the game value considers a worst-case (instead of average) payoff. Our main contributions are as follows.

1. Under natural finiteness assumptions, we show that there exists an optimal strategy for the defender that admits a hidden Markov model and we characterize game payoff for such strategies (Theorem 2.1).

2. We introduce the concept of memory of hidden Markov model that allows to (almost) linearize a highly-nonlinear formula for the game payoff (Theorem 4.1).

3. We introduce SHIELD - an algorithm based on linear programming that computes optimal defender strategies for strategy spaces with a fixed hidden Markov model structure (Section 6).

4. We prove a non-trivial upper bounds for all strategies that admit hidden Markov models (Theorem 5.1).

5. We perform an extensive experimental evaluation of our approach, which includes a computation of a strategy that has 19.3% efficiency against 10.2% that was found in John et al. [2023], *under some additional assumptions* about attacker's behaviour (Section 7).

As a running example, let us consider a USV (an unmanned surface vehicle) that is to patrol the port of Gdynia in Poland (see Figure 1). While introducing the concepts and notation throughout the paper, we will build upon this example to give a better intuition behind the abstract terms.

**Example 1.1.** Figure 1 shows the map of the port of Gdynia in Poland. In 2023 it ranked as the third busiest port in the Baltic Sea in terms of container cargo. The port also features a passenger terminal and is adjacent to the Gdynia Naval Base. Possible routes for the USV are depicted on the map.

## 2 THE MODEL

### 2.1 PATHS IN STATE AND ACTION SPACES

A **state and action space** is a directed graph $(V, E)$ with a set of vertices $V$ that are called **states** and a set of edges $E \subset V \times V$ that are called **actions**. We allow self-loops.

A **path** in a state and action space $(V, E)$ is a sequence $(v_0, v_1, \ldots)$ of states such that $(v_i, v_{i+1}) \in E$. A path may be finite (this includes the **empty path** $\epsilon$) or infinite. We let $V^*$ denote the set of all finite paths and $\mathcal{V}$ denote the set of all infinite paths in $(V, E)$.

For $p \in V^*$, let $|p|$ denote the **length** of $p$ (measured as the length of the sequence of nodes, e.g., a path made of two edges has length 3). For $k \in \mathbb{N}$, we let $V^k \subset V^*$ denote the subset of all paths of length $k$.

For $p \in V^*$ and $q \in V^* \cup \mathcal{V}$ we let $pq = p \cdot q$ denote the **concatenation** of paths $p$ and $q$. If $P \subset V^*$, $Q \subset V^* \cup \mathcal{V}$, then

$$PQ = P \cdot Q = \{pq \colon p \in P, q \in Q, pq \in V^* \cup \mathcal{V}\}.$$

In particular, we let $V^k p = V^k \cdot p$ denote the set of all *paths* that are concatenations of a path of length $k$ with a path $p \in V^*$.

A **shift operator** $\text{shift}_{\mathcal{V}} \colon \mathcal{V} \to \mathcal{V}$ removes the first state from an infinite path $(v_0, v_1, v_2, \ldots)$, i.e.,

$$\text{shift}_{\mathcal{V}}((v_0, v_1, v_2, \ldots)) = (v_1, v_2, \ldots).$$

If $\mathcal{V}$ is known from the context, then we write $\text{shift}$ instead of $\text{shift}_{\mathcal{V}}$.

### 2.2 A GENERAL FORMULATION

Let $(L, R)$ denote a **physical state and action space**. It is a directed graph over which the game is played. We do not make any assumptions about its structure. Elements of $L$ are called **locations** and elements of $R$ are called **routes**. Note that we will commonly use a single element of $L$ to represent a location of multiple patrolling units, cf. Section 3.2.

Let $L^*$ be the set of finite paths in $(L, R)$ which we call **histories**. We let $\epsilon \in L^*$ denote the empty path. Intuitively, a sequence in $L^*$ encodes subsequent positions of the patrolling units during surveillance. We will also interpret elements of $L^*$ to be branches in defender's game tree.

Let $\mathcal{L}$ be the set of infinite paths in $(L, R)$ which we call **patrol schedules** (i.e., we consider the schedules to be extended indefinitely). A **defender strategy** is a probability measure $\mu$ on $\mathcal{L}$. We consider $\mathcal{L}$ to be a measurable space with a $\sigma$-algebra of measurable sets generated by the collection of **cones**:

$$C_p = \{pq \in \mathcal{L} : q \in \mathcal{L}\} \text{ for } p \in L^*.$$

In other words, the cone $C_p$ of $p \in L^*$ is the set of all infinite paths in $\mathcal{L}$ that begin with $p$.

A defender strategy $\mu$ determines how schedules are generated, (cf. Section 2.4.3), so eventually we can think that the defender strategy tells us what is the probability that the patrolling unit will follow a given schedule.

**Example 2.1.** In the case of the port of Gdynia, the physical and action space $(L, R)$ corresponds to the actual physical space of the port, with locations $L$ representing different position in the port area (nodes in the graph on Figure 1), and routes $R$ representing transition routes between these positions (edges in Figure 1). A history in $L^*$ is then a finite path that the USV might take during a daily patrol (probably visiting the same locations multiple times), while $\mathcal{L}$ is the set of patrolling paths that are infinite.

Let $T$ be a finite set of **attack plans** of the attacker. For the sake of generality, we do not impose any additional requirements on $T$ at this moment. It can be, e.g., the set of targets chosen by the attacker (together with the time-lengths of attacks on each target), or the set of paths that the attacker plans to take in order to reach chosen targets.

A **payoff function** is a map $G \colon \mathcal{L} \times T \to \mathbb{R}$. For $j \in T$, we let $G_j = G(\cdot, j)$. We interpret the value of $G_j(p)$ to be the payoff of the defender if the attacker executes an attack plan $j$ at time 0 against the patrol schedule $p \in \mathcal{L}$. Let $n \in \mathbb{N}$ and let $\mathrm{shift}^n$ denote the composition of $\mathrm{shift}$ with itself $n$ times. We let $G_j^n = G_j \circ \mathrm{shift}^n$, $G_j^n \colon \mathcal{L} \to \mathbb{R}$. We interpret $G_j^n(p)$ to be the defender payoff if the attacker executes attack plan $j \in T$ at time $n \in \mathbb{N}$ against the patrol schedule $p \in \mathcal{L}$.

Let $L_+^* = L^* \setminus \{\epsilon\}$. We define **the game value** for strategy $\mu$ of the defender to be

$$V(\mu) = \inf_{i \in L_+^*} \min_{j \in T} \mathrm{E}\left(G_j^{|i|-1} \,\middle|\, C_i\right)$$
$$= \inf_{i \in L_+^*} \min_{j \in T} \frac{1}{\mu(C_i)} \int_{C_i} G_j^{|i|-1} \, \mathrm{d}\mu.$$

Here a pair $(i, j) \in L_+^* \times T$ is an **attacker strategy**, with $i$ being an observation of the attacker (i.e., a sequence of physical states triggering the attack), and $j$ being the attack plan executed as a reaction to observing $i$. Following John et al. [2023], we assume the attack begins at the moment when observation $i$ ends. Hence, the game value depends on the last state of history $i$, and we evaluate $G_j$ after discarding

$|i| - 1$ history states. Therefore, $G_j^{|i|-1} \colon \mathcal{L} \to \mathbb{R}$ is a payoff function of the defender against the strategy $(i, j)$.

**Example 2.2.** In the case of the port of Gdynia, the set of attack plans $T$ can simply be of the set of docks in the port (the red rectangular nodes in Figure 1 are the location of the docks, while the blue round nodes are the non-dock locations) if we assume that the attacker is able to directly reach each dock. However, the attack plans $T$ may also be paths in $(L, R)$ if we assume that the attacker must traverse a path from the port entrance to a given dock. The payoff value $G_j^{\theta-1}(p)$ might express the probability that the attacker launching a strike against target $j$ at time $\theta$ is caught by the USV following a particular patrol route $p$. Similarly, the game value $V(\mu)$ would be the expected probability of apprehending the attacker by a UAV with schedule generated by strategy $\mu$, under assumption that the attacker picks the moment of attack $i$ and the target $j$ optimally.

## 2.3 THE GAME DYNAMICS

Informally, the game defined in Section 2.2 is played between a dynamic defender and a static attacker. The defender is **dynamic** in the sense that they play an extensive-form game (a game with a sequence of moves and incomplete information) over the game tree $L^*$. The attacker is **static** in the sense that they play a normal-form game (a game with a single move and complete information) by picking a single attack plan $j \in T$ to be executed when the defender reaches state $i \in L_+^*$ in the game tree. Note that this distinction is not precise: the set of attack plans $T$ may be very well a set of extensive-form attack strategies transformed into a normal-form with intricacies of player interactions hidden in the definition of the payoff function $G$. Nonetheless, this distinction is important in practice: we assume that sets $L$ and $T$ are not too large, so computations may be performed in a reasonable time.

The definition of the game value $V(\mu)$ accounts for the worst-case scenario for the defender, when the attacker attacks in the worst possible game tree state $i \in L_+^*$. That is, as usual in the security settings, the game value $V(\mu)$ is computed as the *Stackelberg equilibrium*, where the attacker picks their strategy with full knowledge of defender's strategy $\mu$. Note that the infimum $\inf_{i \in L_+^*}$ may be not attained for any $i \in L_+^*$. This is known as **infinitely patient attacker problem** (cf. Vorobeychik et al. [2014]).

The underlying theme of the paper is that the defender patrols some critical infrastructure that is hardly ever attacked. Therefore, attacker's actions and the payoff value are invisible to the defender until the end of the game: the goal is to prevent or deter the attack and we consider the interaction between the defender and the attacker after the attacker is detected to be a separate sub-game that is modeled in a computation of payoff $G$.

## 2.4 FINITENESS ASSUMPTIONS

### 2.4.1 History matching

The usual assumption in the setting of Stackelberg equilibrium is that the attacker knows the mixed strategy of the defender based on the observation of their past actions. It is reasonable to ask: how does the attacker learn defender's strategy? A recent study by Lanctot et al. [2023] has shown that, among many modern machine learning approaches in a similar (albeit simpler) setting, a technique called **history matching** was the most successful.

Let $t \in \mathbb{N}$ be the current moment in time and let $i \in L^{t+1}$. A **context of length** $k$ for some $k \leq t+1$ is a sequence $i_{t-k+1}, i_{t-k+2}, \ldots, i_{t-1}, i_t$ of $k-1$ past states that ends at the current state. History matching tries to match a context to historical data: if $t$ is much greater than $k$, we may look for $t_0 < t$ such that $i_{t_0-j} = i_{t-j}$ for $j = 0, 1, \ldots, k-1$, i.e., a sequence of length $k$ from the past matching the last $k$ states. We want to exploit a possibility that the defender will repeat their past actions if the context matches.

If the defender properly randomizes their actions, the attacker can only exploit statistical data gathered from observation of the past actions in the same context. Since the number of contexts of length $k$ in non-trivial cases grows exponentially with $k$ and since the attacker learns from observation of physical space (which takes time), the **paradigm of the paper** is to assume that the attacker bases their decision about an attack on an observation of a context of length $k$, for some fixed $k$. The notion of observations, on which the attacker bases their decision to execute an attack plan, can be generalized, and its important property is finiteness.

Note that we treat length $k$ of a context observation to be an inherent attribute of attacker's type and we will construct defender's strategy $\mu$ against an assumed value of $k$. This is so because the attacker learns $\mu$ from an observation of defender's action. Alas, it is also a defender's weakness.

### 2.4.2 Actionable observations

The discussion of history matching strategy motivates the following: let $I \subset L_+^*$ be a finite set that we call a set of **actionable observations** for the attacker. We think of actionable observations as of the ones that can trigger an attack decision, i.e., we can say that $i \in L_+^*$ is actionable if upon observing context $i$, the attacker can decide to take action, and otherwise they definitely do not. In other words, after $\theta$ time steps during which the attacker is waiting (absent from the physical state and action space), the attacker observes a sequence $i \in L_+^*$, and if this sequence is actionable, the attacker acts conditionally, according to a chosen attack plan in a way that shall be most beneficial for them.

**Example 2.3.** In the case of the port of Gdynia, an attacker might consider all observation of a given length $k$ as the set of actionable observations. It would correspond to an attacker who records the activities of the USV for a very long time and attempts to predict its route. For $k = 1$ the attacker's reasoning would be: *if the USV is at the moment at dock VII and I start the attack on dock IX, what is the probability that I get caught?* Similarly, for $k = 2$ the reasoning would be: *if the USV is at the moment at dock VII, it arrived from dock VIII, and I start the attack on dock VII, what is the probability that I get caught?* It is worth noting, that as the observation length $k$ increases, the attacker needs to conduct their surveillance for a longer time in order to obtain a reasonable approximation of the defender strategy.

We define the **game value against** $I$ to be

$$V_I(\mu) = \inf_{\theta \in \mathbb{N}} \min_{i \in I} \min_{j \in T} \mathrm{E}\left( G_j^{\theta+|i|-1} \mid C_{L^\theta i} \right).$$

The value $V_I(\mu)$ denotes the value that is the most beneficial for the attacker amongst the choice of an attack plan and an actionable observation. Recall that for $\theta \in \mathbb{N}$, we let $L^\theta$ denote the set of paths of length $\theta$ in $(L, R)$ and let $L^\theta i$ denote the set of all paths from $L^\theta$ concatenated with $i \in L^*$, i.e., $L^\theta i = \{p \cdot i : p \in L^\theta, p \cdot i \in L^*\}$.

In the following, we let $I = L^k$ for $k \in \mathbb{N}$, i.e., we consider attackers that base their attack decision on an observation of a context of length $k$. Note that a setting with $k = 1$ was considered in John et al. [2023] and an arbitrary $k$ was allowed in Basilico et al. [2009]. A distinctive feature of the present paper is to allow the defender to have a hidden state, so $\mu$ may depend on a context that is longer than $k$.

**Lemma 2.1**[1]**.** *We have* $V(\mu) = \inf_{I \subset L_+^*} V_I(\mu)$, *so in particular* $V(\mu) \leq V_I(\mu)$ *for each* $I \subset L_+^*$.

We say that strategy $\mu$ constructed against $I$ is **robust** if $V(\mu) = V_I(\mu)$. Let $\mu^*$ be an **optimal defender strategy** and let $\mu_I^*$ be an **optimal defender strategy against** $I$. Note that an optimal defender strategy against $I$ may be not robust, i.e. it is possible that $V(\mu^*) > V(\mu_I^*)$.

### 2.4.3 Bounded attack resolution time

For $p \in L^*$, we let $P(p) = \mu(C_p)$. It is the probability that the defender will follow history $p$. A **behavioral strategy** of the defender (cf. [Horák et al., 2023, Definition 3.2]) is a map $P(\cdot \mid \cdot) : L \times L^* \to [0, 1]$ defined by the formula $P(q \mid p) = \frac{P(pq)}{P(p)}$ for $p \in L^*$ and $q \in L$, undefined if $P(p) = 0$ or if $pq$ is not a path.

A behavioral strategy $P(\cdot \mid \cdot)$ uniquely determines measure $\mu$ and may be used to sample an element $p \in \mathcal{L}$ according to $\mu$, by recursively sampling the next state $p_{k+1}$ according to the distribution $P(p_{k+1} \mid p_0 p_1 \cdots p_{k-1} p_k)$.

---

[1]Proofs of all theorems are given in the Appendix.

We assume **a bounded attack resolution time**, i.e. that for each $j \in T$ there exists $\tau_j \in \mathbb{N}$ such that for each $p, q \in \mathcal{L}$

if $(p_0, \ldots, p_{\tau_j - 1}) = (q_0, \ldots q_{\tau_j - 1})$, then $G_j(p) = G_j(q)$.

In other words, the payoff $G_j$ depends only on $\tau_j$ initial states of a patrol schedule. In such case we say that an attack plan $j \in T$ **resolves within** $\tau_j$ **turns** (i.e. $\tau_j - 1$ *time steps*). Using this assumption we may define $G_j$ on $L^{\tau_j}$. Let $p \in L^{\tau_j}$. We define $G_j(p)$ to be $G_j(pq)$ for any $q \in \mathcal{L}$ such that $pq \in \mathcal{L}$. We assume that there are no dead ends in $(L, R)$, i.e. that every path can be indefinitely extended.

**Example 2.4.** In the case of the port of Gdynia, the value of $\tau_j$ would correspond to the time necessary to complete the attack on target $j$. For example, the time might be greater for the docks situated deeper into the port if the attacker has to actually traverse the path between the port entrance and their desired target.

**Lemma 2.2.** *Let* $\theta \in \mathbb{N}$, $i \in L_+^*$ *and* $j \in T$ *and let* $\mu$ *be a defender strategy that induces strategy* $P(p) = \mu(C_p)$. *Assume that an attack plan* $j \in T$ *resolves within* $\tau_j$ *turns. We have*

$$
\mathrm{E}\left(G_j^{\theta + |i| - 1} \mid C_{L^\theta i}\right) =
$$
$$
= \frac{1}{\sum_{p \in L^\theta i} P(p)} \sum_{p \in L^\theta i L^{\tau_j - 1}} P(p) G_j^{\theta + |i| - 1}(p).
$$

### 2.4.4 Time-invariance

Intuitively, a defender's strategy that depends on time (or in other words - a behavioral strategy that depends on the depth in the game tree) may be vulnerable to a properly timed attack. In the definition of $V_I(\mu)$, the attacker picks a time of attack $\theta \in \mathbb{N}$ that is most favorable to him. Therefore in the present paper we restrict our attention to defender strategies $\mu$ that are shift-**invariant** or **time-invariant**, i.e. strategies such that $\mu(L \cdot A) = \mu(A)$ for every measurable set $A \subset \mathcal{L}$ and for the set of locations $L$. An important example of time-invariant measure is a push-forward of a Markov measure that is introduced in Section 4.1.

**Example 2.5.** In the case of the port of Gdynia, a shift-invariant defender strategy would select the next destination of the USV based on the finite number of previous actions, without looking indefinitely far into the past.

We say defender strategy $\mu$ is **discrete** if the range of the behavioral strategy, $\{P(s \mid p) : p \in L^*, s \in L\}$, is finite. Otherwise we say that $\mu$ is **continuous**. Note that for each $I \subset L_+^*$ and for each discrete defender strategy $\mu$ there exists a time-invariant strategy $\nu$ such that $V_I(\nu) \geq V_I(\mu)$. However, we do not know if every continuous defender strategy may be approximated by a discrete strategy with a close game value against $I$.

If strategy $\mu$ is shift-invariant, then $V_I(\mu)$ can be computed by the following formula that involves only a finite set of parameters and a minimization over a finite set.

**Lemma 2.3.** *Let* $\theta \in \mathbb{N}$, $i \in L^*$ *and* $j \in T$ *and let* $\mu$ *be a defender strategy that induces strategy* $P(p) = \mu(C_p)$. *Assume that an attack plan* $j \in T$ *resolves within* $\tau_j$ *turns. If* $\mu$ *is* shift-*invariant, then*

$$
\mathrm{E}\left(G_j^{\theta + |i| - 1} \mid C_{L^\theta i}\right) = \frac{1}{P(i)} \sum_{p \in i L^{\tau_j - 1}} P(p) G_j^{|i| - 1}(p).
$$

**Definition 2.1.** For $i \in L_+^*$, $p \in L^*$ we let

$$
P(i \sim p) = \begin{cases} \frac{P(i \operatorname{shift}(p))}{P(i)} & \text{if } p_0 = i_{|i| - 1}, \\ 0 & \text{if } p_0 \neq i_{|i| - 1}. \end{cases}
$$

Using $\theta$ to denote $|i| - 1$, $P(i \sim p)$ is a conditional probability that the patrol schedule from time $\theta$ is equal to $p$ under the condition that from time $0$ it equals $i$. Observe that these intervals overlap: the last state of $i$ has to be equal to the first state of $p$, otherwise the probability is $0$. Note that $P(i \sim p)$ is undefined if $P(i) = 0$.

**Theorem 2.1.** *Assume that an attack plan* $j \in T$ *resolves within* $\tau_j$ *turns. If* $\mu$ *is* shift-*invariant, then the game value against* $I \subset L_+^*$ *is equal to*

$$
V_I(\mu) = \min_{i \in I} \min_{j \in T} \sum_{p \in L^{\tau_j}} P(i \sim p) G_j(p).
$$

## 3 A PATROLLING GAME

In this section, we describe a specific method of constructing a physical space $(L, R)$, attack plans $T$ and payoff functions $G_j$. What gets constructed this way is a **patrolling game** which is a crucial instance of our general model.

### 3.1 A PATROLLING SETTING

A patrolling setting (environment) is a model of protecting critical infrastructure viewed statically, before any dynamic interplay between the defender and the attacker is taken into account. The setting consists of the following:

1. A set $U$ of patrolling units, a set $T$ of protected targets (corresponding to the attack plans), and a set of defender's values of targets $V : T \to \mathbb{R}$.

2. A directed graph $(L_u, R_u)$ defined for each patrolling unit $u \in U$. The graph describes the topology of the critical infrastructure that we protect, and consists of: the set of locations that are being patrolled $L_u$, and the set of connecting routes (edges) $R_u \subset L_u \times L_u$.

   We allow self-loops in $R_u$. Each route has its length $d_u : R_u \to \mathbb{N}_+$. The lengths can vary between the

edges, and are specified in time units. We can think of them as depending both on physical distance between locations and on speed of the patrolling unit.

3. A coverage function $\Gamma_u : L_u \times T \to [0, 1]$ defined for each patrolling unit $u \in U$. For a patrolled location $l \in L_u$ and a target $t \in T$, the function $\Gamma_u(l, t)$ is the probability that the patrolling unit $u$ stationed at location $l$ will catch an intruder within a single unit of time while he attacks target $t$.

It is easier to understand the coverage function when it is binary-valued. Then, for each unit $u$ the set $\{t \in T : \Gamma_u(l, t) = 1\}$ can be interpreted as the targets protected by the unit $u$ from the vertex $l$. We generalize this notion to take into account the possibility of imperfect target protection.

**Example 3.1.** In the case of the port of Gdynia, the set $U$ might consist of a USV and a UAV (unmanned aerial vehicle). These two patrolling units might then have different patrolling routes, thus different $(L_u, R_u)$ graphs (e.g., the UAV could fly directly between any two locations, while the USV can only travel on water). The coverage function could express the fact that a patrolling unit situated at location $l$ corresponding to a given dock $t$ fully protects it ($\Gamma_u(l, t) = 1$), while also partially protecting dock $t'$ whose corresponding location is connected to $l$ with an edge ($\Gamma_u(l, t') = \frac{1}{2}$). The value $V$ might be greater for military docks, and smaller for the civilian ones.

## 3.2 A PHYSICAL STATE AND ACTION SPACE

A **physical space** $(L, R)$ (mentioned in Section 2.2), representing the dynamics of the defensive force, provides a unified framework where a single state uniquely represents an arrangement of multiple defensive resources and each action takes a single unit of time. We construct $(L, R)$ and a coverage function $\Gamma : L \times T \to [0, 1]$, using patrolling environment data specified in Section 3.1.

First, for each unit $u \in U$, we get rid of the length function $d_u$ by subdividing long edges of the graph $(L_u, R_u)$ into several edges of unit length. The procedure is detailed in Appendix A.1. We extend the coverage function $\Gamma_u$ to intermediate vertices by setting coverage to 0, i.e. no target is protected when unit is in an intermediate state. Note that any other extension would work with our method, e.g. a linear interpolation of coverage from both ends of the long edge.

Then, the physical space $(L, R)$ is defined to be a tensor product of subdivided graphs. A coverage function $\Gamma : L \times T \to [0, 1]$ is defined by the formula

$$\Gamma(v, t) = 1 - \prod_{u \in U} (1 - \Gamma_u(\pi_u(v), t)),$$

where $\pi_u : L \to L_u$ denotes a projection from the product graph onto its $u$-th factor, i.e., selecting the location of

the unit $u$ from the vector of all unit locations. While in our formula we assume that each patrolling unit has an independent chance to catch the intruder, any other joint distribution of coverage functions would work.

**Example 3.2.** In the case of the port of Gdynia, the states $L$ of the physical space could correspond to the pairs of the position of the USV and the position of the UAV, while the actions $R$ to the transitions of both units to new positions. Assume that each unit provides coverage 1 for the location where it is positioned, and $\frac{1}{2}$ to adjacent locations. The coverage function $\Gamma((l_1, l_2), t)$ would then have the value of 1 for dock $t$ corresponding to either $l_1$ or $l_2$, $\frac{3}{4}$ for docks adjacent to both $l_1$ and $l_2$, $\frac{1}{2}$ for docks adjacent to either $l_1$ or $l_2$ but not both, and 0 for all other docks.

We can think of the physical space, $(L, R)$, as a board on which a game between the defender and the attacker is played. We consider a **static** attacker who commits to a single decision to perform an attack on a target $j \in T$. Let $\tau : T \to \mathbb{N}_+$ denote the **attack duration** of targets. Once the attacker commits to attack $j \in T$, the defender has $\tau_j$ turns to catch them. The defender patrols the locations according to a patrolling schedule $p \in \mathcal{L}$. The probability that the defender will successfully defend target $j \in T$ is:

$$D_j(p) = 1 - \prod_{t=0}^{\tau_j - 1} \left(1 - \Gamma(p_t, j)\right).$$

The formula is based on an assumption that at each moment of time the patrolling units have independent chance to capture the attacker. Like before, this assumption is not essential and the method presented in the paper will work with any joint distribution. Let $V : T \to \mathbb{R}$ denote the **value** of targets, equal for both players. We assume that the game is constant-sum and this assumption is essential in our paper. Defender's payoff depends both on $p$ and $j$ and is equal to

$$G_j(p) = V(j)D_j(p).$$

# 4 A STRATEGY SPACE

## 4.1 A HIDDEN MARKOV MODEL

The key idea of our work is to let the defender's hidden state be more elaborate than the position of their patrolling units, thus allowing a more complex behavior and giving advantage against certain types of attackers. To this end, we let $(S, A)$ be a **strategy state and action space**, used by the defender to control their defensive resources. We equip $(S, A)$ with a graph homomorphism $X : S \to L$, called **projection**. We use $X$ to translate defender's actions in the strategy space (internal scheduling) to actions in the physical space $(L, R)$ (movement of patrolling units).

To explain the paradigm of a hidden Markov model let us consider an example shown in Figure 2. The physical space

has 4 locations connected into a star-shaped graph with a center node $x$ and three leaves $T = \{a, b, c\}$ that are possible targets of attack. There is a single patrolling unit. Number of turns to attack each target is equal to 4, $\tau_a = \tau_b = \tau_c = 4$. Each target has value 1 and the patrolling unit protects only the node that it is visiting. Consider an attacker with observation length 1, i.e. consider a set $I = \{a, b, c, x\}$ of actionable observations. We look for a strategy $\mu$ that maximizes $V_I(\mu)$.

First, consider a case where the patrolling unit moves from the center node $x$ to each of leaf nodes $a, b, c$ with uniform probability $\frac{1}{3}$. This is a Markov chain model that was considered previously in the literature [John et al., 2023]. One of optimal attacker strategies against it is to attack node $c$ when the patrolling unit is observed at node $a$; the probability that the attack will be intercepted is then equal to $\frac{1}{3}$.

A better strategy for the defender against this type of attacker is to never visit the same leaf node twice in a row: if the patrolling unit arrived at position $x$ from leaf $a$, then it should move either to $b$ or to $c$ with probability $\frac{1}{2}$. An optimal attacker strategy does not change, but the probability that the attack will be intercepted increases to $\frac{1}{2}$. Note that this strategy is robust: the payoff will remain the same if we add elements to the set of actionable observations. The reason behind this is that even if we increase the observation length of the attacker, allowing them to distinguish between the strategy states of the defender, they cannot avoid capture with probability greater than $\frac{1}{2}$. Their set of strategies is then: attacking the peripheral node that the defender just left, attacking one of the other peripheral nodes than the one that the defender just left, or attack one of the other peripheral nodes than the one that the defender is at right now. All of these strategies result in getting captured with probability $\frac{1}{2}$. Therefore, by Lemma 2.1, this is an optimal strategy for the defender.

This strategy may be realized by a Markov chain on a strategy space $(S, A)$ (see Figure 2). The projection $X$ translates actions in the strategy space into actions in the physical space $(L, R)$, i.e., into the movements of the patrolling unit.

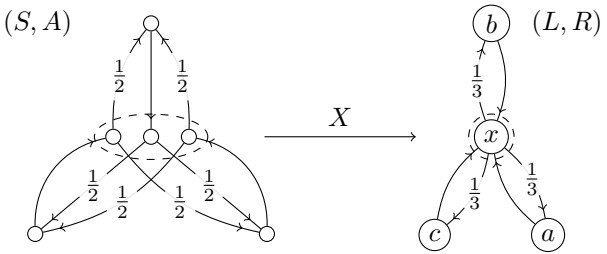

Figure 2: A strategy space (left) over a star-graph with three leaves (right). The hidden states over the center allow for construction of more sophisticated strategy that increases a payoff from $\frac{1}{3}$ to $\frac{1}{2}$ against an opponent making attack decision based on the position of the single patrolling unit.

To define a hidden Markov model, let $\mathcal{M}$ be a Markov chain on $(S, A)$ with a *transition matrix* $N$ and a *stationary distribution* $\sigma$. Let $\mathcal{S}$ be a set of infinite paths in $(S, A)$, let $\nu$ be a *Markov measure* induced by $\mathcal{M}$ on $\mathcal{S}$ (cf. [Sarig, 2009, Definition 1.8]), and let $\mu = \nu \circ X^{-1}$ be a *push-forward* of measure $\nu$ from $\mathcal{S}$ to $\mathcal{L}$. Measure $\mu$ is a defender strategy and we call $\nu$ a **hidden Markov model** for $\mu$.

Note that by [Sarig, 2009, Proposition 1.8] the Markov measure $\nu$ is $\text{shift}_{\mathcal{S}}$-invariant, so its push-forward $\mu = \nu \circ X^{-1}$ is $\text{shift}_{\mathcal{L}}$-invariant. Thus, Theorem 2.1 applies to defender strategies with hidden Markov models. The following lemma relates measure $\mu = \nu \circ X^{-1}$ to the transition probabilities $N$ and the stationary distribution $\sigma$ of the hidden Markov chain.

**Lemma 4.1.** *Assume that $\mu$ has a hidden Markov model with a stationary distribution $\sigma$, transition matrix $N$ and projection $X \colon S \to L$. Let $X_* \colon S^* \to L^*$ be a natural map induced by $X$, i.e., the element-wise application of $X$. Then for each $p \in L^*$ we have*

$$P(p) = \mu(C_p) = \sum_{q \in X_*^{-1}(p)} \sigma_{q_0} \prod_{i=0}^{|p|-2} N_{q_i, q_{i+1}}.$$

### 4.2 A SPACE WITH MEMORY

In this section, we introduce the key concept of the paper – **state and action spaces with memory**. Let $(S, A)$ be a strategy state and action space. Let $Z$ be a function on $S$ with arbitrary range $\text{rg}(Z)$. We say that $(S, A)$ has **memory of length $t$ with respect to $Z$** (where $t \geq 1$) if the following condition holds for each pair of states $r, s \in S$:

> if $Z(r) \neq Z(s)$, then for each pair of paths $rr_1 r_2 r_3 \cdots r_{t-1}$, $ss_1 s_2 s_3 \cdots s_{t-1}$ of length $t$ we have $r_{t-1} \neq s_{t-1}$.

In other words, if $Z$ differentiates states $r$ and $s$, then the internal defender's state after any $t - 1$ actions will still be different. The principle might be easier to understand in its contrapositive form: for all $r, s \in S$, if for some paths $rr_1 r_2 r_3 \cdots r_{t-1}$ and $ss_1 s_2 s_3 \cdots s_{t-1}$ we have $r_{t-1} = s_{t-1}$, then $Z(r) = Z(s)$. Thus, the current internal state $s \in S$ of the defender determines uniquely what attacker's observation was $t - 1$ steps ago. In other words, the states in $S$ contain the information about the last $t$ observations of the attacker. Note that it does not force the state space to be large, as is shown in Appendix D, together with a couple of other methods and examples of constructions of spaces with memory.

Let $M_Z \colon S \times \{0, \ldots, t-1\} \to \text{rg}(Z)$, where $\text{rg}(Z)$ denotes the range of $Z$, be a **memory function** that maps a pair $(s, i)$ to the past value of $Z$, i.e., its value $i$ steps before the defender reached the state $s$. Lemma 4.2 shows that $M_Z$ is properly defined.

**Lemma 4.2.** *Assume that the $(S, A)$ has memory of length $t$ with respect to $Z$ for $t \geq 1$ and that each $s \in S$ has at least one incoming edge. Then the memory function $M_Z \colon S \times \{0, \dots, t-1\} \to \mathrm{rg}(Z)$ that satisfies*

*for each $s \in S$ and each $p \in S^t$ such that $p_{t-1} = s$*
*we have $M_Z(s, i) = Z(p_{t-1-i})$*

*is well-defined.*

## 4.3 CONSTRUCTING SPACES WITH MEMORY

In our approach, a strategy state and action space is a space with memory of length $t$ with respect to projection $X$. Such spaces may be constructed in several ways. See Appendix D for a more detailed discussion.

The most straightforward approach is to construct a **space of paths**, where each state is a path of length $t$ in the original physical graph. Such a space may be endowed with additional states: a tensor product of a space with memory $t$ with an arbitrary graph is again a space with memory $t$. A space of paths may be filtered by heuristics, e.g., we may consider only simple paths as elements of the strategy space.

Interestingly, a number of states in a space with memory $t$ doesn't have to be large, as is seen in the construction of **space of disjoint cycles**. Such a space replicates the usual approach to patrolling with Stackelberg games in matrix form, cf. [Shieh et al., 2012].

## 4.4 A LIFT OF ATTACKER'S OBSERVATION

We assumed that attacks are triggered by histories sampled from $L^* i$, where $i \in I$ is an actionable observation. We now assume that the actionable observations are the sequences of length $h$ from $L$, i.e., $I = L^h$ and we let $h \in \mathbb{N}$ be an **observation length**.

Let $X$ be a projection from $(S, A)$ to $(L, R)$ defined in Section 4.1. Let $Y_h \colon L^* \cdot L^h \to L^h$ defined by

$$Y_h(p) = (p_{|p|-h}, \dots, p_{|p|-2}, p_{|p|-1})$$

be a **context of length** $h$ of history $p \in L^*$ such that $|p| \geq h$. Intuitively, $Y_h(p)$ selects the last $h$ elements of $p$.

If the strategy space $(S, A)$ has a memory of length at least $h$ with respect to $X$, then we may **lift** the observation $Y_h$ to be a function of $s \in S$:

$$\widetilde{Y}_h(s) = (M_X(s, h-1), M_X(s, h-2), \dots, M_X(s, 0)),$$

where $M_X$ is the memory function defined in Section 4.2. Intuitively, $\widetilde{Y}_h(s)$ produces the last $h$ states of the physical space $(L, R)$ based on the current state $s$ of the strategy space $(S, A)$.

Consider $s \in S^*$, a sequence of internal strategy states. Let $p \in L^*$ be the result of applying projection $X$ to each element of $s$, i.e., $p = X_*(s)$ using notation introduced in Lemma 4.1. Then $Y_h(p) = \widetilde{Y}_h(s_{|s|-1})$, i.e., the history of length $h$ in the physical space is encoded by the *last* state $s_{|s|-1}$ in the strategy space.

Motivated by the above property, we define an **attacker's observation function** to be a function $Y \colon S \to I$, where $I = L^h$ is the attacker's set of actionable observations. Notice the assumption that the strategy space $(S, A)$ has memory at least $h$, i.e., the internal space of the defender is rich enough to encompass the attacker's actionable observations.

Note that the dependence of $Y$ on $S$ does not mean that the attacker observes the internal state of the defender. It means that the strategy state space $S$ is complex enough to reconstruct the attacker's observation. This assumption is very useful from the technical point of view, as it simplifies the description of the model and its solution.

## 4.5 A SWITCH OF PERSPECTIVE

We now introduce a crucial formula that switches perspective from the future into the past, allowing us to (almost) linearize a highly non-linear formula given in Theorem 2.1 and Lemma 4.1.

**Lemma 4.3.** *Let $i \in L^*$ and $t \in \mathbb{N}$. Let*

$$\hat{H}_{i,t} = \left\{ s \in S \colon \left( X_*^{-1}(i) \cdot S^t \right) \cap (S^* \cdot s) \neq \emptyset \right\}$$

*be a set of all states in strategy space that are reachable after following path $i$ in the physical space and continuing for $t$ time steps. If $(S, A)$ has a memory of length $|i| + t$ with respect to $X$ and $P$ has a hidden Markov model with stationary distribution $\sigma$, then $P(i) = \sum_{s \in \hat{H}_{i,t}} \sigma_s$.*

Assume that the strategy space $(S, A)$ has memory of length at least $\tau_j$ with respect to $X$ and $Y$, where $\tau_j$ is the duration of the attack plan $j$, while $X$ and $Y$ are defined in Section 4.4. Notice that the payoff function $G_j$ of the attack plan $j$ depends only on the last $\tau_j$ states of the physical space. Hence, we can lift the payoff function $G_j$:

$$\widetilde{G}_j(s) = G_j(M_X(s, \tau_j-1), M_X(s, \tau_j-2), \dots, M_X(s, 0)),$$

where $\widetilde{G}_j(s) \colon S \to \mathbb{R}$, and $M_X$ is a memory function. Moreover, for $i \in I$ let

$$H_{i,t} = \{s \in S \colon M_Y(s, t) = i\},$$

i.e., $H_{i,t}$ is the set of states where $t$ time units ago the attacker's observation returned an actionable observation $i$.

**Theorem 4.1.** *Assume that $(S, A)$ has memory of length $\max_{j \in T} \tau_j$ with respect to $X$ and $Y$, and that the defender*

*strategy $\mu$ has a hidden Markov model with a stationary distribution $\sigma$. Then*

$$V_I(\mu) = \min_{i \in I} \min_{j \in T} \frac{\sum_{s \in H_{i,\tau_j - 1}} \sigma_s \widetilde{G}_j(s)}{\sum_{s \in H_{i,\tau_j - 1}} \sigma_s}.$$

## 5 AN UPPER BOUND THEOREM

Now we prove an upper bound theorem for patrolling games introduced in Section 3. As a corollary we obtain a method of computing upper bounds via the linear problem 1, which vastly generalizes methods that exist in the literature.

**Theorem 5.1.** *If $\mu$ is* shift-*invariant, then*

$$V(\mu) \le \min_{j \in T} V(j) \tau_j \sum_{s \in L} P(s) \Gamma(s, j).$$

Note that if $\mu$ has a hidden Markov model, then probabilities $P(s)$ satisfy network-flow conditions on graph $(L, R)$. Therefore the following linear program computes an upper bound on $V(\mu)$ for any strategy $\mu$ that admits a hidden Markov model.

$$
\begin{aligned}
\max_{\xi, \sigma, N} \quad & \xi \\
\text{s.t.} \quad & \sum_{w \in L} \sigma_w = 1 \\
& \sum_{v \in L: (w,v) \in R} N_{w,v} = \sigma_w \text{ for } w \in L \\
& \sum_{v \in L: (v,w) \in R} N_{v,w} = \sigma_u \text{ for } w \in L \quad (1) \\
& \xi \le V(j) \tau_j \sum_{s \in L} \sigma_s \Gamma(s, j) \text{ for } j \in T \\
& \xi \in \mathbb{R} \\
& \sigma_w \in [0, 1] \text{ for } w \in L \\
& N_{w,v} \in [0, 1] \text{ for } (w, v) \in R
\end{aligned}
$$

## 6 SHIELD

In this section we introduce **Security Heuristic for Intrusion Exposure and Location Defense (SHIELD)** – an algorithm that constructs a nearly optimal hidden Markov model strategy for the defender for a patrolling game with a fixed set of actionable observations of the attacker. The setup used for the algorithm is identical with one used in Section 4.5.

First, we construct a space $(S, A)$ with memory of length $\max_{i \in I} |i| + \max_{j \in T} \tau_j - 1$. Although any such space works, the choice restricts the set of available strategies $\mu$, so the game value $V_I(\mu)$ depends on this choice.

By Theorem 4.1, the goal of the defender is to find the stationary distribution $\sigma$ of a Markov chain on $(S, A)$ with

---

**Algorithm 1** Approximating $V^*$ using the bisection method.

**Input:** Strategy space $(S, A)$, set of actionable observations $I$, set of attack plans $T$, defender payoff function $G$, $\epsilon > 0$.
**Output:** The lower and upper bound of $V^*$ precise up to $\epsilon$.
1: $V_L^* \leftarrow 0$
2: $V_U^* \leftarrow \max_{j \in T, v \in S} G_j(v)$
3: **while** $V_U^* - V_L^* > \epsilon$ **do**
4:      $V_M^* \leftarrow (V_U^* + V_L^*)/2$
5:      **if** linear problem 2 feasible with $\xi = V_M^*$ **then**
6:          $V_L^* \leftarrow V_M^*$
7:      **else**
8:          $V_U^* \leftarrow V_M^*$
9: **return** $V_L^*, V_U^*$

---

the maximal value $V^* = \max_\sigma V_I(\mu)$. Thus, if we fix $\xi \in \mathbb{R}$, the following linear problem is feasible iff $\xi \le V^*$.

$$
\begin{aligned}
\max_{\sigma, N} \quad & 0 \\
\text{s.t.} \quad & \sum_{w \in S} \sigma_w = 1 \\
& \sum_{v \in S: (w,v) \in A} N_{w,v} = \sigma_w \text{ for } w \in S \\
& \sum_{v \in S: (v,w) \in A} N_{v,w} = \sigma_w \text{ for } w \in S \quad (2) \\
& \sum_{w \in H_{i,\tau_j - 1}} \sigma_w \left( \tilde{G}_j(w) - \xi \right) \ge 0, i \in I, j \in T \\
& \sigma_w \in [0, 1] \text{ for } w \in S \\
& N_{w,v} \in [0, 1] \text{ for } (w, v) \in A
\end{aligned}
$$

Having the above linear formulation, we can approximate the value of $V^*$ arbitrarily well by using the bisection method (see pseudocode in Algorithm 1).

**Example 6.1.** To finish our example of the port of Gdynia, consider the network presented in Figure 1 with all docks being equally valuable to the defender, and all non-docks being worthless. Assume that each USV provides coverage 1 for dock corresponding to the node where it is positioned, and coverage $\frac{1}{2}$ to adjacent docks. Moreover, assume that it takes 3 time units to attack each of the docks, and the actionable observations of the attacker are all sequences of length 1, i.e., the attacker makes their decision based on the current positions of the USVs. In such a setting, the optimal probability of capturing the attacker calculated by our algorithm is $0.25$ if the defender has one USV at their disposal, and it grows to $0.95$ if we add another USV.

## 7 EXPERIMENTAL EVALUATION

We evaluate our solution on real-life and random networks. Our implementation is published at github.com/anagorko/stackelberg-games-core.

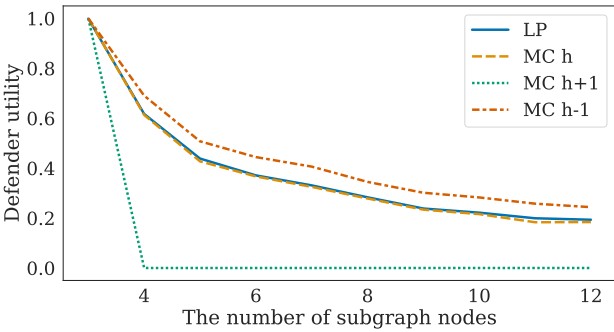

Figure 3: The defender utility in subgraphs of the San Francisco network. Each line corresponds either to value computed either via our linear program (LP) or via Monte Carlo simulations (MC) with different values of attacker's observation length. Each Mone Carlo data point is an average over $10^3$ roll-outs with $10^3$ actions each. The colored areas (extremely narrow) represent $95\%$ c.i.

## 7.1 SAN FRANCISCO POLICE DISTRICT

In the model by John et al. [2023], the defender controls a single patrol unit, and the set of targets consists of twelve intersections in downtown San Francisco. The targets are connected into a weighted clique with integer weights representing the minutes of travel time between intersections, ranging from 1 to 9, with attack times of the targets ranging from 6 to 11. The attacker observes the current position of the security unit, i.e., the observation length is 1. The authors of John et al. [2023] use a JAX-based gradient optimizer to find a patrolling solution with utility equal to 0.102. Our algorithm is able to identify a patrolling solution with the lower bound of the defender utility equal to 0.193. **In other words, we are able to find a defense strategy where the probability of apprehending the attacker is almost two times greater than the state of the art**.

## 7.2 SENSITIVITY TO OBSERVATION LENGTH

Our algorithm computes an optimal strategy against an attacker with a given observation length $h$. However, it remains unclear how would such strategy fare against an attacker with other observation lengths. To investigate it we now run Monte Carlo simulations on increasingly large induced subgraphs of the San Francisco network. For each subgraph, we generate $10^3$ roll-outs of the defender strategy consisting of $10^3$ actions each. We then assume that the attack has either observation length $h$, $h+1$, or $h-1$, and they select the target and observation from the roll-out that yield the smallest average risk of getting caught, i.e., the greatest utility of the attacker. Figure 3 presents the results. The theoretical bound computed by the linear program is confirmed by our simulations. Moreover, the strategy is even more successful against an opponent with shorter observation length.

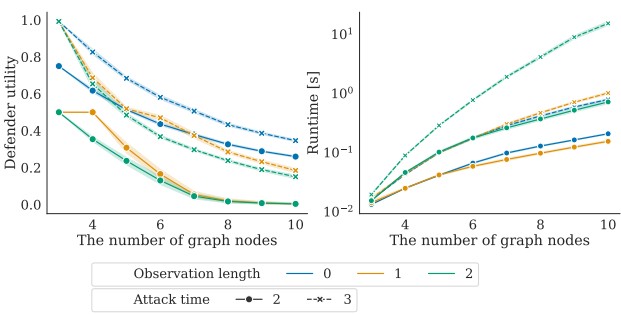

Figure 4: The left plot presents the mean utility of the defender, while the right one the mean runtime. Each data point is an average over 100 Barabási-Albert networks. The colored areas (very narrow) represent $95\%$ c.i.

Unfortunately, an attacker with longer observation length is able to capitalize on the strategy optimized against a weaker opponent and inevitably avoids detection.

## 7.3 EVALUATION ON RANDOM NETWORKS

To evaluate the effects of the network size and structure on the outcomes of our experiments, we also perform simulations with randomly networks generated. To this end, we use Barabási-Albert 1999, Erdős-Rényi 1959, and Watts-Strogatz 1998 models. We generate networks of varying size, while setting the average degree of a node to 2. In the case of the Watts-Strogatz model we set the rewiring probability to $\frac{1}{4}$. For each such network we calculate the utility of the defender using a single security resource. All simulations in this section are run on a computer with Intel Core i7-11700K CPU, and 16 GB RAM. Figure 4 presents the results of our simulations for the Barabási-Albert, the results for the other two models can be found in the supplementary materials, and exhibit similar trends. As can be seen both increasing the observation length of the attacker, as well as decreasing the attack time of the nodes can significantly lower the utility of the defender. In particular, an attacker with observation length zero becomes more dangerous if we give them the ability to observe the defender's activities than if we decrease their attack time. Unfortunately, increasing the observation length results in a sharp growth of the run time required to compute the optimal strategy, exacerbating the danger posed by a well-informed attacker.

## 8 CONCLUSIONS

In this work, we proposed a model on the interface of stochastic patrolling and game theoretic models. We constructed an effective algorithm and showed that it improved upon state of the art for some settings in the literature.

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

# General Markov Model for Solving Patrolling Games
## (Supplementary Material)

**Andrzej Nagórko**[1,2]       **Marcin Waniek**[1,2]       **Małgorzata Róg**[1]

**Michał Godziszewski**[1,2]       **Barbara Rosiak**[1]       **Tomasz P. Michalak**[1,2]

[1]IDEAS NCBR, Warsaw, Poland
[2]Faculty of Mathematics, Informatics and Mechanics, University of Warsaw, Warsaw, Poland

## A   AUXILIARY DEFINITIONS

### A.1   LONG EDGE SUBDIVISION

Long edge subdivision used in Section 3.2 is described as follows.

Given a route $r \in R_u$ connecting two vertices $l_i$ and $l_j$ such that $|r| > 1$, we add $|r| - 1$ intermediate vertices $l_1^r, \ldots, l_{|r|-1}^r$ between the nodes $l_i$ and $l_j$ and connect them by new edges, each of length 1, i.e., instead of having a long edge $r = (l_i, l_j) \in R_u$ we now have the following $|r|$ directed short edges (i.e., each of length 1): $r_0 = (l_i, l_1^r)$, $r_1 = (l_1^r, l_2^r)$, ..., $r_k = (l_k^r, l_{k+1}^r)$, ..., $r_{|r|-1} = (l_{|r|-1}^r, l_j)$.

This way, we obtain a graph $\mathcal{T}_u' = (L_u', R_u')$ with broken down edges, where

$$L_u' = L_u \cup \bigcup_{r \in R_u : |r| > 1} \{l_k^r : k = 1, \ldots, |r| - 1\},$$

and

$$R_u' = \big(R_u \setminus \{r \in R_u : |r| > 1\}\big) \cup$$
$$\cup \bigcup_{r = (l_i, l_j) \in R_u : |r| > 1} \{(l_i, l_1^r), (l_1^r, l_2^r), \ldots, (l_{|r|-1}^r, l_j)\}.$$

### A.2   TENSOR PRODUCT OF GRAPHS

Given two graphs $G = (V_G, E_G)$ and $H = (V_H, E_H)$, the tensor product $G \times H$ of these graphs is defined as follows: the set of vertices $V_{G \times H}$ is equal to the cartesian product $V_G \times V_H$ of the sets of vertices of $G$ and $H$, and a pair $((v_0, u_0)(v_1, u_1))$ is an edge of $G \times H$ iff $(v_0, v_1) \in E_G$, and $(u_0, u_1) \in E_H$.

# B PROOFS

## B.1 THE PROOF OF LEMMA 2.1

**Lemma 2.1.** *We have $V(\mu) = \inf_{I \subset L_+^*} V_I(\mu)$, so in particular $V(\mu) \leq V_I(\mu)$ for each $I \subset L_+^*$.*

*Proof.* From the definition of $V_I(\mu)$,

$$V_I(\mu) = \inf_{\theta \in \mathbb{N}} \min_{i \in I} \min_{j \in T} \mathrm{E}\left(G_j^{\theta+|i|-1} \,\Big|\, C_{L^\theta i}\right).$$

By setting $I = \{i\}$ for $i \in L_+^*$ and $\theta = 0$, we obtain

$$\inf_{I \subset L_+^*} V_I(\mu) \leq \inf_{i \in L_+^*} V_{\{i\}}(\mu) \leq \inf_{i \in L_+^*} \min_{j \in T} \mathrm{E}\left(G_j^{0+|i|-1} \,\Big|\, C_{L^0 i}\right) = V(\mu),$$

with the last equality coming from the definition of $V(\mu)$. Hence

$$\inf_{I \subset L_+^*} V_I(\mu) \leq V(\mu).$$

To prove the reverse inequality, first observe that $C_{L^\theta i} = \bigsqcup_{p \in L^\theta} C_{pi}$ for each $i \in I$ and $\theta \in \mathbb{N}$, where $\bigsqcup$ denotes disjoint union. Hence, by the law of total expectation,

$$\mathrm{E}\left(G_j^{\theta+|i|-1} \,\Big|\, C_{L^\theta i}\right) = \sum_{p \in L^\theta} \frac{\mu(C_{pi})}{\mu(C_{L^\theta i})} \mathrm{E}(G_j^{\theta+|i|-1} \mid C_{pi}).$$

Since $\sum_{p \in L^\theta} \frac{\mu(C_{pi})}{\mu(C_{L^\theta i})} = 1$, we have

$$\sum_{p \in L^\theta} \frac{\mu(C_{pi})}{\mu(C_{L^\theta i})} \mathrm{E}\left(G_j^{\theta+|i|-1} \,\Big|\, C_{pi}\right) \geq \min_{p \in L^\theta} \mathrm{E}\left(G_j^{\theta+|i|-1} \,\Big|\, C_{pi}\right).$$

Finally,

$$V(\mu) = \inf_{i \in L_+^*} \min_{j \in T} \mathrm{E}\left(G_j^{|i|-1} \,\Big|\, C_i\right) =$$

$$= \inf_{\theta \in \mathbb{N}} \min_{p \in L^\theta} \inf_{i \in L_+^*} \min_{j \in T} \mathrm{E}(G_j^{|pi|-1} \mid C_{pi}) =$$

$$= \inf_{\theta \in \mathbb{N}} \inf_{i \in L_+^*} \min_{j \in T} \left(\min_{p \in L^\theta} \mathrm{E}\left(G_j^{\theta+|i|-1} \,\Big|\, C_{pi}\right)\right) \leq$$

$$\leq \inf_{\theta \in \mathbb{N}} \inf_{i \in I} \min_{j \in T} \mathrm{E}\left(G_j^{\theta+|i|-1} \,\Big|\, C_{L^\theta i}\right) = V_I(\mu).$$

$\square$

## B.2 A VALUE OF THE GAME – GENERAL FORMULATION

**Lemma 2.2.** *Let $\theta \in \mathbb{N}$, $i \in L_+^*$ and $j \in T$ and let $\mu$ be a defender strategy that induces strategy $P(p) = \mu(C_p)$. Assume that an attack plan $j \in T$ resolves within $\tau_j$ turns. We have*

$$\mathrm{E}\left(G_j^{\theta+|i|-1} \,\Big|\, C_{L^\theta i}\right) =$$

$$= \frac{1}{\sum_{p \in L^\theta i} P(p)} \sum_{p \in L^\theta i L^{\tau_j - 1}} P(p) G_j^{\theta+|i|-1}(p).$$

*Proof.* From the definition of conditional expected value,

$$\mathrm{E}\left(G_j^{\theta+|i|-1} \,\middle|\, C_{L^\theta i}\right) = \frac{1}{\mu(C_{L^\theta i})} \int_{C_{L^\theta i}} G_j^{\theta+|i|-1} \,\mathrm{d}\mu.$$

Hence, we have to prove that:

1. $\mu(C_{L^\theta i}) = \sum\limits_{p \in L^\theta i} P(p),$

2. $\int_{C_{L^\theta i}} G_j^{\theta+|i|-1} \,\mathrm{d}\mu = \sum\limits_{p \in L^\theta i L^{\tau_j-1}} P(p) G_j^{\theta+|i|-1}(p).$

From the definition of a cone, we have

$$C_{L^\theta i} = \bigsqcup_{q \in L^\theta} C_{qi} = \bigsqcup_{q \in L^\theta} \bigsqcup_{r \in L^{\tau_j-1}} C_{qir},$$

where $\bigsqcup$ denotes a disjoint union (notice that $p$ iterates over all possible sequences of the length $\tau_j - 1$ that can be the extensions of $qi$).

Therefore, from the definition of $P$ we have

$$\mu(C_{L^\theta i}) = \mu\left(\bigsqcup_{q \in L^\theta} C_{qi}\right) = \sum_{q \in L^\theta} \mu(C_{qi}) = \sum_{q \in L^\theta} P(qi) = \sum_{p \in L^\theta i} P(p)$$

which completes the proof of the first point.

Moreover, we have

$$\int_{C_{L^\theta i}} G_j^{\theta+|i|-1} \,\mathrm{d}\mu = \sum_{q \in L^\theta} \sum_{r \in L^{\tau_j-1}} \int_{C_{qir}} G_j^{\theta+|i|-1} \,\mathrm{d}\mu =$$

$$= \sum_{q \in L^\theta} \sum_{r \in L^{\tau_j-1}} \mu(C_{qir}) G_j^{\theta+|i|-1}(qir) =$$

$$= \sum_{q \in L^\theta} \sum_{r \in L^{\tau_j-1}} P(qir) G_j^{\theta+|i|-1}(qir) =$$

$$= \sum_{p \in L^\theta i L^{\tau_j-1}} P(p) G_j^{\theta+|i|-1}(p),$$

since $G_j^{\theta+|i|-1}$ is constant on $C_{qir}$ and equal to $G_j^{\theta+|i|-1}(qir)$ by the assumption that attack plan $j$ resolves within $\tau_j$ turns. This completes the proof of the second point, and the Lemma. $\square$

## B.3   A VALUE OF THE GAME – SHIFT-INVARIANT STRATEGY

**Lemma 2.3.** *Let $\theta \in \mathbb{N}$, $i \in L^*$ and $j \in T$ and let $\mu$ be a defender strategy that induces strategy $P(p) = \mu(C_p)$. Assume that an attack plan $j \in T$ resolves within $\tau_j$ turns. If $\mu$ is* shift*-invariant, then*

$$\mathrm{E}\left(G_j^{\theta+|i|-1} \,\middle|\, C_{L^\theta i}\right) = \frac{1}{P(i)} \sum_{p \in i L^{\tau_j-1}} P(p) G_j^{|i|-1}(p).$$

*Proof.* From the definition of conditional expected value,

$$\mathrm{E}\left(G_j^{\theta+|i|-1} \,\middle|\, C_{L^\theta i}\right) = \frac{1}{\mu(C_{L^\theta i})} \int_{C_{L^\theta i}} G_j^{\theta+|i|-1} \,\mathrm{d}\mu.$$

From shift-invariance of $\mu$, i.e. $\mu = \mu \circ \mathrm{shift}^{-1}$, we have

$$\mu(C_{L^\theta i}) = \mu(L^\theta C_i) = \mu(C_i) = P(i).$$

Using integration by substitution,

$$\int_{C_{L^\theta i}} G_j^{\theta+|i|-1} \, \mathrm{d}\mu = \int_{\mathrm{shift}^{-\theta}(C_i)} G_j^{|i|-1} \circ \mathrm{shift}^\theta \, \mathrm{d}\mu = \int_{C_i} G_j^{|i|-1} \, \mathrm{d}\mu.$$

$\square$

## B.4   THE PROOF OF THEOREM 2.1

**Theorem 2.1.** *Assume that an attack plan $j \in T$ resolves within $\tau_j$ turns. If $\mu$ is shift-invariant, then the game value against $I \subset L_+^*$ is equal to*

$$V_I(\mu) = \min_{i \in I} \min_{j \in T} \sum_{p \in L^{\tau_j}} P(i \sim p) G_j(p).$$

*Proof.* From the definition of $V_I(\mu)$,

$$V_I(\mu) = \inf_{\theta \in \mathbb{N}} \min_{i \in I} \min_{j \in T} \mathrm{E}\left(G_j^{\theta+|i|-1} \,\Big|\, C_{L^\theta i}\right).$$

By Lemma 2.3,

$$\mathrm{E}\left(G_j^{\theta+|i|-1} \,\Big|\, C_{L^\theta i}\right) = \frac{1}{P(i)} \sum_{p \in iL^{\tau_j-1}} P(p) G_j^{|i|-1}(p).$$

Hence

$$V_I(\mu) = \inf_{\theta \in \mathbb{N}} \min_{i \in I} \min_{j \in T} \frac{1}{P(i)} \sum_{p \in iL^{\tau_j-1}} P(p) G_j^{|i|-1}(p) =$$

$$= \min_{i \in I} \min_{j \in T} \sum_{p \in iL^{\tau_j-1}} \frac{P(p)}{P(i)} G_j^{|i|-1}(p) =$$

$$= \min_{i \in I} \min_{j \in T} \sum_{p \in L^{\tau_j-1}} \frac{P(ip)}{P(i)} G_j^{|i|-1}(ip) =$$

$$= \min_{i \in I} \min_{j \in T} \sum_{p \in L^{\tau_j}} P(i \sim p) G_j(p).$$

$\square$

## B.5   A NON-LINEAR FORMULATION FOR PROBABILITY OF FOLLOWING A PATH

**Lemma 4.1.** *Assume that $\mu$ has a hidden Markov model with a stationary distribution $\sigma$, transition matrix $N$ and projection $X\colon S \to L$. Let $X_*\colon S^* \to L^*$ be a natural map induced by $X$, i.e., the element-wise application of $X$. Then for each $p \in L^*$ we have*

$$P(p) = \mu(C_p) = \sum_{q \in X_*^{-1}(p)} \sigma_{q_0} \prod_{i=0}^{|p|-2} N_{q_i, q_{i+1}}.$$

*Proof.* Let $q \in S^*$ such that $X_*(q) = p$. From the definition of a Markov measure [Sarig, 2009, Definition 1.8], we have

$$\nu(C_q) = \sigma_{q_0} \prod_{i=0}^{|p|-2} N_{q_i, q_{i+1}}.$$

We have $X_*^{-1}(C_p) = \bigsqcup_{q \in X_*^{-1}(p)} C_q$. where $\bigsqcup$ denotes a disjoint union. From the definition of a push-forward measure,

$$\mu(C_p) = (\nu \circ X^{-1})(C_p) = \nu(X_*^{-1}(C_p)) =$$

$$= \nu \left( \bigsqcup_{q \in X_*^{-1}(p)} C_q \right) = \sum_{q \in X_*^{-1}(p)} \sigma_{q_0} \prod_{i=0}^{|p|-2} N_{q_i, q_{i+1}}.$$

$\square$

## B.6  EXISTENCE OF A MEMORY FUNCTION

**Lemma 4.2.** *Assume that the $(S, A)$ has memory of length $t$ with respect to $Z$ for $t \geq 1$ and that each $s \in S$ has at least one incoming edge. Then the memory function $M_Z \colon S \times \{0, \ldots, t-1\} \to \mathrm{rg}(Z)$ that satisfies*

$$\text{for each } s \in S \text{ and each } p \in S^t \text{ such that } p_{t-1} = s$$
$$\text{we have } M_Z(s, i) = Z(p_{t-1-i})$$

*is well-defined.*

*Proof.* Fix the space $(S, A)$, the observation function $Z$ and assume the space has memory of length $t$ w.r.t. $Z$. We need to demonstrate that for every $i \leq t$ the value $M_Z(s, i)$ is well-defined. Indeed, fix $i \leq t-1$ and fix $s \in S$. Then, since the space has the memory of length $t$ with respect to $Z$, by looking at the contrapositive reading of the condition defining the memory length it is trivial to note that the observation $Z(p_{t-1-i})$ is uniquely determined for any sequence of actions $a_1, \ldots, a_i \in A$ leading from $p_{t-1-i}$ to $s$. Therefore, the definition $M_Z(s, i) := Z(p_{t-1-i})$ is correct. $\square$

## B.7  THE PROOF OF LEMMA 4.3

**Lemma 4.3.** *Let $i \in L^*$ and $t \in \mathbb{N}$. Let*

$$\hat{H}_{i,t} = \left\{ s \in S \colon \left( X_*^{-1}(i) \cdot S^t \right) \cap (S^* \cdot s) \neq \emptyset \right\}$$

*be a set of all states in strategy space that are reachable after following path $i$ in the physical space and continuing for $t$ time steps. If $(S, A)$ has a memory of length $|i| + t$ with respect to $X$ and $P$ has a hidden Markov model with stationary distribution $\sigma$, then $P(i) = \sum_{s \in \hat{H}_{i,t}} \sigma_s$.*

*Proof.* From the definition, we have

$$P(i) = \mu(C_i) = \nu(X_*^{-1}(C_i)) = \nu(X_*^{-1}(i) \cdot S^t \cdot \mathcal{S}),$$

where $\nu$ is a hidden Markov model for $\mu$. Observe that from the assumption that $S$ has a memory of length $|i| + t$ with respect to $X$, we have

$$\hat{H}_{i,t} \cap \hat{H}_{j,t} = \emptyset \text{ for all } j \in L^{|i|} \text{ such that } i \neq j.$$

It follows that

$$X^{-1}(i) \cdot S^t = \left\{ q \in S^{|i|+t} \colon q_{|i|+t} \in \hat{H}_{i,t} \right\} = S^{|i|+t-1} \cdot \hat{H}_{i,t}.$$

Therefore,

$$\nu(X_*^{-1}(i) \cdot S^t \cdot \mathcal{S}) = \nu(S^{|i|+t-1} \cdot \hat{H}_{i,t} \cdot \mathcal{S}) = \nu(\hat{H}_{i,t} \cdot \mathcal{S}),$$

the last equality from shift-invariance of $\nu$. Finally, from additivity of $\nu$ and from Lemma 4.1, we have

$$\nu(\hat{H}_{i,t} \cdot \mathcal{S}) = \sum_{s \in \hat{H}_{i,t}} P(C_s) = \sum_{s \in H_{i,t}} \sigma_s.$$

$\square$

## B.8   THE PROOF OF THEOREM 4.1

**Theorem 4.1.** *Assume that $(S, A)$ has memory of length $\max_{j \in T} \tau_j$ with respect to $X$ and $Y$, and that the defender strategy $\mu$ has a hidden Markov model with a stationary distribution $\sigma$. Then*

$$V_I(\mu) = \min_{i \in I} \min_{j \in T} \frac{\sum_{s \in H_{i, \tau_j - 1}} \sigma_s \widetilde{G}_j(s)}{\sum_{s \in H_{i, \tau_j - 1}} \sigma_s}.$$

*Proof.* We have $i \in I \subset L^*$ and

$$H_{i, \tau_j - 1} = \{s \in S \colon M_Y(s, \tau_j - 1) = i\} =$$
$$= \left\{s \in S \colon M_X(s, \tau_j - 1 + k) = i_{|i| - 1 - k} \text{ for } k = 0, 1, \ldots, |i| - 1\right\} =$$
$$= \left\{s \in S \colon \left(X_*^{-1}(i) \cdot S^{\tau_j - 1}\right) \cap (S^* \cdot s) \neq \emptyset\right\} = \hat{H}_{i, \tau_j - 1}.$$

Hence by Lemma 4.3, we have $P(i) = \sum_{s \in H_{i, \tau_j - 1}} \sigma_s$. Recall that

$$\widetilde{G}_j(s) = G_j(M_X(s, \tau_j - 1), M_X(s, \tau_j - 2), \ldots, M_X(s, 0)).$$

From Theorem 2.1, we have

$$V_I(\mu) = \min_{i \in I} \min_{j \in T} \sum_{p \in L^{\tau_j}} P(i \sim p) G_j(p) =$$
$$= \min_{i \in I} \min_{j \in T} \sum_{p \in i_{|i| - 1} L^{\tau_j - 1}} \frac{P(i \, \mathrm{shift}(p))}{P(i)} G_j(p) =$$
$$= \min_{i \in I} \min_{j \in T} \frac{\sum_{p \in i_{|i| - 1} L^{\tau_j - 1}} P(i \, \mathrm{shift}(p)) G_j(p)}{\sum_{s \in H_{i, \tau_j - 1}} \sigma_s}.$$

Note that $\widetilde{G}_j$ is constant on $\hat{H}_{i \, \mathrm{shift}(p), 0}$ since $(S, A)$ has a memory of length $\tau_j$ with respect to $X$. We also have

$$H_{i, \tau_j - 1} = \bigsqcup_{p \in i_{|i| - 1} L^{\tau_j - 1}} \hat{H}_{i \, \mathrm{shift}(p), 0}.$$

Hence

$$\sum_{p \in i_{|i| - 1} L^{\tau_j - 1}} P(i \, \mathrm{shift}(p)) G_j(p) = \sum_{p \in i_{|i| - 1} L^{\tau_j - 1}} \left(\sum_{s \in \hat{H}_{i \, \mathrm{shift}(p), 0}} \sigma_s\right) G_j(p) =$$
$$= \sum_{p \in i_{|i| - 1} L^{\tau_j - 1}} \left(\sum_{s \in \hat{H}_{i \, \mathrm{shift}(p), 0}} \sigma_s \tilde{G}_j(s)\right) = \sum_{s \in H_{i, j}} \sigma_s \widetilde{G}_j(s).$$

$\square$

## B.9   PROOF OF THEOREM 5.1

**Theorem 5.1.** *If $\mu$ is* shift-*invariant, then*

$$V(\mu) \leq \min_{j \in T} V(j) \tau_j \sum_{s \in L} P(s) \Gamma(s, j).$$

*Proof.* Directly from the definition we have

$$V(\mu) \le \min_{j \in T} \min_{i \in L} \mathrm{E}\left(G_j \mid C_i\right).$$

Since the attack plan $j$ resolves in $\tau_j$ turns, we have (cf. proof of Lemma 2.3)

$$\mathrm{E}\left(G_j \mid C_i\right) = \frac{1}{P(i)} \sum_{p \in iL^{\tau_j - 1}} P(p) G_j(p).$$

Since $\sum_{i \in L} P(i) = 1$, we have

$$\min_{i \in L} \sum_{p \in iL^{\tau_j - 1}} \frac{P(p)}{P(i)} G_j(p) \le \sum_{i \in L} P(i) \sum_{p \in iL^{\tau_j - 1}} \frac{P(p)}{P(i)} G_j(p) = \sum_{p \in L^{\tau_j}} P(p) G_j(p).$$

Hence

$$V(\mu) \le \min_{j \in T} \sum_{p \in L^{\tau_j}} P(p) G_j(p).$$

Let's notice that:

$$D_j(p) = 1 - \prod_{t=0}^{\tau_j - 1} \left(1 - \Gamma(p_t, j)\right) \le \sum_{t=0}^{\tau_j - 1} \Gamma(p_t, j),$$

and

$$\sum_{p \in L^{\tau_j}} P(p) \left( \sum_{t=0}^{\tau_j - 1} \Gamma(p_t, j) \right) = \sum_{t=0}^{\tau_j - 1} \sum_{p \in L^{\tau_j}} \Gamma(p_t, j) P(p) \le \sum_{s \in L} \Gamma(s, j) \left[ \sum_{t=0}^{\tau_j - 1} \sum_{p \in L^{\tau_j}} \delta_{p_t = s} P(p) \right],$$

where $\delta$ is the Kronecker delta (i.e., $\delta_{p_t = s} = 1$ if $p_t = s$, and $\delta_{p_t = s} = 0$ otherwise). Since $\mu$ is shift-invariant, for each $s \in L$ and each $t$ we have

$$\sum_{p \in L^{\tau_j}} \delta_{p_t = s} P(p) = P(s),$$

hence

$$\sum_{p \in L^{\tau_j}} P(p) \left( \sum_{t=0}^{\tau_j - 1} \Gamma(p_t, j) \right) \le \tau_j \sum_{s \in L} \Gamma(s, j) P(s).$$

Therefore

$$V(\mu) \le \min_{j \in T} \sum_{p \in L^{\tau_j}} P(p) G_j(p)$$

$$= \min_{j \in T} \sum_{p \in L^{\tau_j}} P(p) \left( 1 - \prod_{t=0}^{\tau_j - 1} (1 - \Gamma(p_t, j)) \right) V(j)$$

$$= \min_{j \in T} V(j) \sum_{p \in L^{\tau_j}} P(p) \left( 1 - \prod_{t=0}^{\tau_j - 1} (1 - \Gamma(p_t, j)) \right)$$

$$\le \min_{j \in T} V(j) \sum_{p \in L^{\tau_j}} P(p) \left( \sum_{t=0}^{\tau_j - 1} \Gamma(p_t, j) \right)$$

$$\le \min_{j \in T} V(j) \tau_j \sum_{s \in L} P(s) \Gamma(s, j).$$

$$\square$$

## C EXAMPLES

### C.1 THE MODEL FROM John et al. [2023] AS AN INSTANCE OF OUR GENERAL MODEL

Let us describe the framework by John et al. [2023] that analyzes stochastic surveillance strategies of randomized patrolling robots, using the formalism of our model. Let $H = (L, R)$ be a physical state and action space. Assume that a Markov chain $P$ is given and that the defender patrols $H$ according to $P$, and let $\mu_P$ be the defender strategy derived from $P$. Consider the set of attack plans $T$ to be a subset of $L$ (where each target $l_j \in T$ implicitly contains the information on the attack-time $\tau_j$ of the vertex $l_j$). The set of strategies of the attacker is

$$\{(\lambda, l_j) : \lambda \in L^*, l_j \in L\},$$

where $\lambda$ is a (finite) sequence of nodes of $L$ visited by the patrolling unit until the moment of the attack. If we let $X_k$ to be the value of $P$ at time $k$, i.e., the node visited at time $k$, then

$$t_{ij} = \min\{k : X_0 = l_i, X_k = l_j\}$$

is a random variable representing the number of time periods between the agent leaving the node $l_i \in L$ and their arrival to node $l_j \in L$. The payoff of the defender $G_j(p)$ is equal to 1, if $t_{ij} \leq \tau_j$, and $-1$ otherwise, where $p_0 = l_i$, i.e., if at the time 0 of the schedule $p$ the defender is located in the node $l_i$. The game value of the strategy $\mu$ of the defender is then simply

$$V(\mu) = \min_{l_i \in L} \min_{j \in T} \mu \left( t_{ij} \leq \tau_j \right).$$

### C.2 A STAR GRAPH

Consider a physical and strategy states shown in Figure 5. There is a single patrolling unit. The set of targets is the set of vertices of the physical space and the attack time for each vertex is equal to 3. We assume that we are playing against an opponent with observation length 1, i.e. an opponent that observes the current position of the patrolling unit.

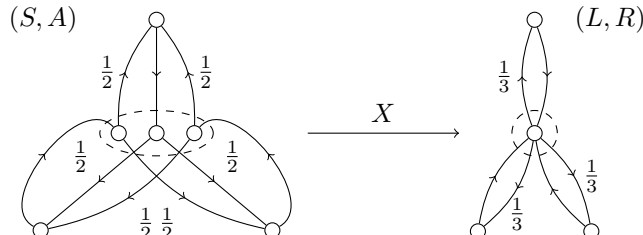

Figure 5: A strategy space (left) over a star-graph with three leaves (right). The hidden states over the center allow for construction of more sophisticated strategy, as described in Section C.

We consider two defense strategies: (1) the patrolling unit is governed by a Markov chain defined on the physical space; (2) the patrolling a Markov chain defined on the strategy space. In both cases the optimal Markov chain selects its actions with uniform probability. However, the expected payoff for the defender playing strategy (1) is $\frac{1}{3}$ and it increases to $\frac{1}{2}$ when he switches to strategy (2).

### C.3 A 5-CYCLE

Consider a 5-cycle as presented in Figure 6. Let all nodes be targets with the same value and attack time equal 2.

Let us consider an attacker with memory of length 2. According to results from exact solver, defender's strategy with hidden states is able to achieve up to $\frac{1}{3}$ capture probability, as compared to $\frac{1}{4}$ for one without memory. An example of optimal strategy for the defender is moving in sequences of two edges clockwise or two edges counterclockwise. After a sequence, he should randomly choose the direction of the next by picking the same as previously with probability $\frac{1}{3}$ and reverse with probability $\frac{2}{3}$.

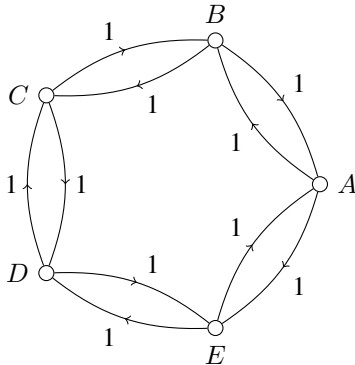

Figure 6: A physical space based on a 5-cycle.

Notice that the attacker will always benefit from attacking target two edges away from defender's current position. Assume that defender's last move was from $E$ to $A$. Then the probability that defender will move to $B$ and then to $C$ is the probability that when he finishes the sequence, he chooses to continue going counterclockwise, which is $\frac{1}{3}$. The probability that defender will move to $E$ and then to $D$ is the probability that he just finished sequence of two moves and now starts going clockwise, which is $\frac{1}{2} \cdot \frac{2}{3} = \frac{1}{3}$. This shows that capture probability is $\frac{1}{3}$.

## C.4 A $2n+1$-CYCLE

The above strategy can actually be generalized. Let $(L, R)$ be a cycle with $2n+1$ vertices. Assume all of them are targets with the same value and the attack time is equal to $n$.

As above, suppose the attacker has memory of length 2. Consider the following strategy: make $n$ steps in one direction (randomly choosing the clockwise direction or the anti-clockwise one), and then randomly choose the direction of the next sequence of $n$ moves by drawing the same direction as for the previous sequence with probability $\frac{1}{n+1}$ and reverse the direction with probability $1 - \frac{1}{n+1}$.

Again, the attacker benefits most from attacking a target that is $n$ edges away from defender's current location. Thus, by the same reasoning as before, this strategy is quaranteed to give the defender the capture probabiltiy equal to $\frac{1}{n+1}$ against a rational attacker.

## C.5 COMPARISON WITH GAME VALUES COMPUTED BY Horák et al. [2023]

We can compare game values of the model from Horák et al. [2023] to the ones we compute with our model. Over there, the game value is defined as:

$$V_I(\mu) = \inf_\sigma \sum_{i=1}^\infty \gamma^{i-1} P_{b,\mu,\sigma}(s^{(i)}) R(s^{(i)} a_1^{(i)} a_2^{(i)}),$$

where $s^i$ is the $i$-th state of the game (in our terminology: the $i$-th element of the patrol schedule), $\sigma$ is the attacker strategy, $P_{b,\mu,\sigma}$ is the probability of the state in the $i$-th round of the game being $s$, depending on the initial distribution $b$, and the players strategies, $a_j^i$ are actions of $j$-th player in the $i$-th round, and $R$ are payoffs of the defender for a given round. If we apply similar finiteness and invariance assumptions, as we have above, to this model in Horák et al. [2023], set discount factor $\gamma = 1$, assume that actions of attacker do not affect the defender, but that his activity is implicit in the objective function, and replace the sum of $R$-s with $G_j(p)$ then the game value from Horák et al. [2023] becomes in the notation of our model:

$$V_I(\mu) = \inf_{j \in T} \sum_{i \in I} \sum_{p \in L^{\tau_j}} P(ip) G_j(p).$$

That means that we compute game value as a worst case scenario, while Horák et al. [2023] computes game value as an average. We believe that in security scenarios the former is more adequate.

# D  CONSTRUCTING SPACES WITH MEMORY

There are many ways of constructing strategy spaces with memory. For instance, given a physical space $(L, R)$, and a positive integer $k$, we may define the states of $S$ to be $k + 1$-tuples of nodes from $L$, interpreted as as the current position of the patrolling unit, together with $k$ positions visited immediately before. Then, $|S| = L^{k+1}$, and $S$ has memory of length $k$ with respect to $X$. For $m \leq k$ the attacker can then observe the current position of the unit and its previous $m$ locations.[1] This construction is illustrated in the section D.1 of the Appendix.

Consider another example of a strategy space with memory. Given an arbitrary physical space $(L, R)$, we can construct the strategy space as the set of mutually disjoint finite cycles $\{C_{k_i} : i = 1, \ldots, |L|\}$, each of length $k_i$. Such a space is equivalent to selecting a mixed strategy in a Stackelberg game where pure strategies are patrols from a predetermined set, similarly to the models built in recent years in many widely applicable works on security games Sinha et al. [2018]. Since in such a space almost all moves of the defender but the initial one are deterministic, the memory of the space (with respect to $X$) is actually infinite, despite the fact that the size of the space can be relatively small.

Further, consider constructing the strategy space by taking a tensor product of the physical space $(L, R)$ and an arbitrary graph $G$. In particular, the graph $G$ might be a clique, in which case it can be seen as an internal memory of the defender. Aside from knowing their location in the physical space, the defender can use $G$ to store an additional piece of information, with the number of distinct states equal to the number of nodes in $G$. The length of memory then depends on $G$, notice however that as a result of the tensor multiplication the length of the memory cannot decrease.

## D.1  SPACE WITH MEMORY OVER THE 5-CYCLE

To see a concrete example of a specific construction of space with memory, the physical space $(L, R)$ to be the cycle graph $C_5$ with 5 vertices $\{l_0, \ldots, l_4\}$, and edges in both directions, i.e.,

$$R = \{(l_i, l_{i+1}), (l_i, l_{i-1}) : i = 0, \ldots, 4\},$$

where the addition and subtraction are defined modulo 5. Suppose all edges have equal length 1, and that each vertex stores a target of equal positive value. Now assume we have one patrolling unit and define $S := L^3$, i.e., the states of $S$ are triples of vertices, interpreted as the current position $l$ of the unit, together with the two vertices visited by the unit immediately before $l$, where if e.g.,

$$s_i = (l_1, l_3, l_0),$$

then it means the current position of the patrol is $l_1$, to which it arrived from $l_3$, and one step earlier it was in $l_0$. In other words, the states of the strategy space can be identified with paths of length 2. Obviously, $|S| = 125$. We may define the attacker's observation $Y : S \to \mathbb{Y}$ in such a way that $\mathbb{Y} = L^2$, for any

$$s_i = (l_{i_1}, l_{i_2}, l_{i_3}) \in S$$

define

$$Y(s_i) = (l_{i_1}, l_{i_2}),$$

that is the attacker observes the current position of the patrolling unit and its previous location (the attacker, observing the physical space, remembers the location from which the patrolling unit came to the current position). The formal requirement for this to be well-defined is for the space has memory of length 1 with respect to $X$. The space $S$ has actually memory equal to the length of the paths. Assume now that the sequence of physical states of the patrol is the repeated cycle $l_0, l_1, l_2, l_3, l_4$. Then, in the strategy space, the sequence of internal strategy space generated by the defender will be

$$s = ((l_0, l_4, l_3), (l_1, l_0, l_4), (l_2, l_1, l_0), \ldots).$$

Then, the attacker's observations will be

$$((l_0, l_4), (l_1, l_0), (l_2, l_1), \ldots).$$

This is an instance of a strategy space, where each node can be identified with with a path of (given fixed length) nodes from the physical space.

It is worthwhile to observe in this place that if the strategy space $S$ is a cyclic graph (even when we forget about the direction of the edges), then if for a given natural number $n$, the girth of the graph, i.e., the length of a shortest cycle contained in the graph, is greater or equal than $2n$, then the space has memory at least $n$ (it can be actually larger).

---

[1]The formal requirement for this to be well defined is for the space to have memory of length $m - 1$ with respect to $X$.

# E SUPPLEMENTARY FIGURES

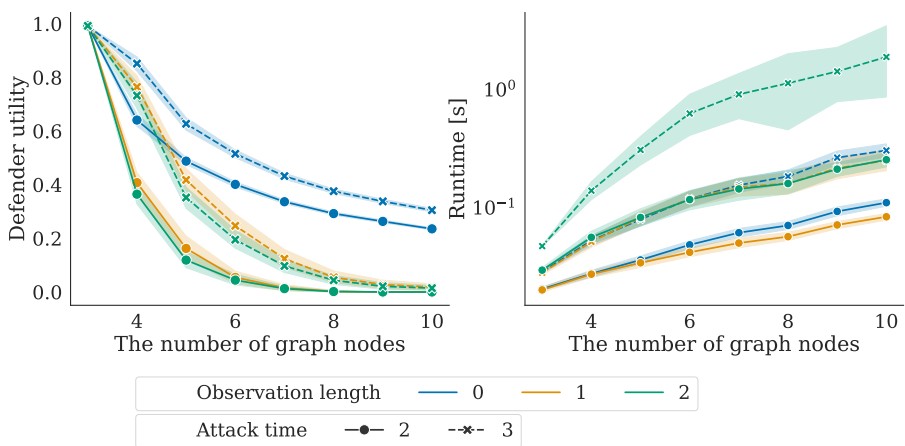

Figure 7: The left plot presents the mean utility of the defender, while the right plot presents the mean runtime. Each data point is an average over 100 Erdős-Rényi networks. The colored areas (very narrow in most cases) represent $95\%$ confidence intervals.

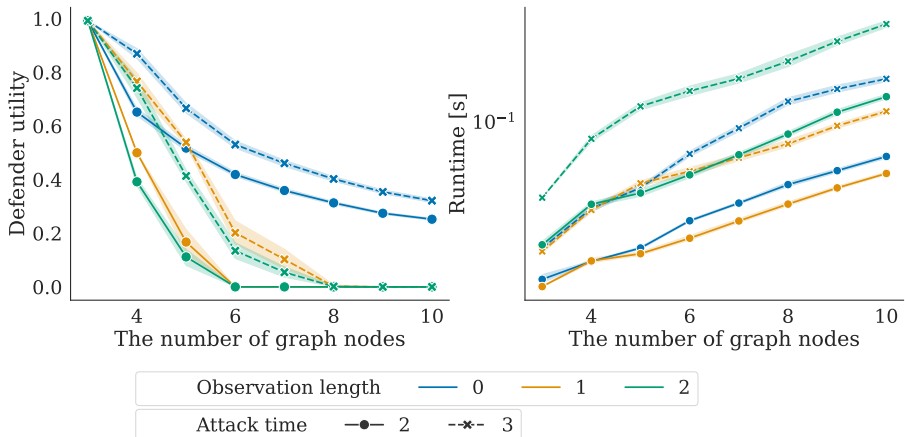

Figure 8: The left plot presents the mean utility of the defender, while the right plot presents the mean runtime. Each data point is an average over 100 Watts-Strogatz networks. The colored areas (very narrow) represent $95\%$ confidence intervals.

# F GLOSSARY

| Notation | Description |
|---|---|
| $C_p$ | A cone in $\mathcal{L}$ over $p \in L^*$, i.e. $$C_p = \{q \in \mathcal{L}\colon p_i = q_i \text{ for } i \leq |p|\}.$$ |
| $G$ | The payoff function of the defender $G\colon \mathcal{L} \times T \to \mathbb{R}$, where $G(p,j)$ is the payoff of the defender if the attacker executes an attack plan $j$ at time 0 against the patrol schedule $p$. |
| $\Gamma_u$ | Coverage function $\Gamma_u\colon L_u \times T \to [0,1]$ defined for each patrolling unit $u \in U$, where $\Gamma_u(l,t)$ is the probability that the patrolling unit $u$ stationed at location $l$ will catch an intruder within a single unit of time while he attacks target $t$. |
| $G_j$ | The payoff function of the defender when attacker executes attack plan $j \in T$, i.e., $G_j = G(\cdot, j)$. |
| $\widetilde{G}_j$ | Lift of the payoff function, i.e., $\widetilde{G}_j(s) = G_j((M_X(s, \tau_j - 1), M_X(s, \tau_j - 2), \ldots, M_X(s, 0)))$, where $M_X(s,i)$ maps $(s,i)$ to the past value of $X$ from $i$ steps before the defender reached the state $s$. |
| $G_j^n$ | The defender payoff if the attacker executes attack plan $j \in T$ at time $n \in \mathbb{N}$ against the patrol schedule $p \in \mathcal{L}$, i.e. $$G_j^n = G_j \circ \text{shift}^n.$$ |
| $I$ | Actionable observations of the attacker, i.e. a set $I \subset L^*$, i.e., upon observing $i \in I$, the attacker can decide to take action, and otherwise they definitely does not make motion to perform an attack plan. |
| $(i,j)$ | An attacker strategy, i.e., a pair $(i,j) \in L^* \times T$ of a moment of the attack $i \in \mathbb{N}$ and an attack plan $j \in T$. |
| $\mathcal{L}$ | A set of infinite paths in $(L,R)$, i.e. $$\mathcal{L} = \{(s_1, s_2, s_3, \ldots)\colon s_i \in L, (s_i, s_{i+1}) \in R\}.$$ |
| $L^*$ | A set of finite paths in a state and action space $(L,R)$, i.e. $$L^* = \{p = (p_1, p_2, \ldots, p_k)\colon p_i \in L, (p_i, p_{i+1}) \in R\}.$$ We allow an empty path, for $k = 0$, denoted $\epsilon$. |
| $(L,R)$ | A state and action space, i.e. a directed graph, $R \subset L \times L$. |
| $(L_u, R_u)$ | The set of locations $L_u$ patrolled by a particular unit $u \in U$, and the set of routes $R_u$ that it traverses. |
| $\mathcal{M}$ | A Markov chain on $(S,A)$ with a transition matrix $N$ and a stationary distribution $\sigma$. |
| $\mu$ | The defender strategy $\mu = \nu \circ X^{-1}$ with a hidden Markov model $\nu$. |
| $\mu^*$ | An optimal defender strategy, i.e. $$\mu^* \in \arg\max_P V(\mu).$$ |
| $\mu_I^*$ | An optimal defender strategy against $I$, i.e. $$\mu_I^* \in \arg\max_\mu V_I(\mu).$$ |
| $N$ | Transition matrix of the Markov chain. |
| $\nu$ | Markov measure induced by $\mathcal{M}$ on $\mathcal{S}$. |

| Notation | Description |
|---|---|
| $P$ | Behavioral strategy of the defender, i.e., $P\left(s \mid p\right)$ is the probability that the defender enters location $s \in L$, given that so far they have been following the patrolling schedule $p$. |
| $\|p\|$ | For $p = (p_1, p_2, \ldots, p_k) \in L^*$, we let $\|p\| = k$ denote the length of $p$. |
| $p \cdot q$ | If $p, q \in L^*$, then $p \cdot q$ is concatenation of paths, i.e. $$p \cdot q = (p_1, p_2, \ldots, p_k, q_1, q_2, \ldots, q_l),$$ where $p = (p_1, p_2, \ldots, p_k)$ and $q = (q_1, q_2, \ldots, q_l)$. If we write $p \cdot q$, then we implicitly assume that $p \cdot q \in L^*$, i.e. $(p_k, q_1) \in R$. |
| $\mathcal{S}$ | A set of infinite paths in $(S, A)$. |
| $(S, A)$ | Strategy state and action space used by the defender to control their defensive resources. |
| shift | A shift operator $\mathrm{shift} \colon \mathcal{L} \to \mathcal{L}$, i.e. $$\mathrm{shift}(s_1, s_2, s_3, \ldots) = (s_2, s_3, \ldots).$$ |
| $\sigma$ | Stationary distribution of the Markov chain. |
| $T$ | The set of attack plans of the attacker. |
| $\tau_j$ | The attack duration of attack plan $j \in T$. |
| $U$ | The set of patrolling units. |
| $V(\mu)$ | Game value against the defender strategy $\mu$, i.e. $$V(\mu) = \inf_{I \subset L^*} V_I(\mu).$$ |
| $V$ | Defender's values of targets $V \colon T \to \mathbb{R}$. |
| $V_I$ | Game value against the set of actionable observations $I$, i.e., the value that is most beneficial for the attacker amongst the choice of an attack plan and an actionable observation. |
| $X$ | Projection, i.e., a graph homomorphism $X \colon S \to L$ that translates defender's internal scheduling to actions in the physical space. |
| $Y$ | Attacker's observation function $Y \colon S \to I$. |
| $Y_h$ | History of length $h$, i.e., $Y_h(p) = \mathrm{shift}^{\|p\|-h}(p)$. |
| $\widetilde{Y}_h$ | Lift of the attacker's observation function, i.e., $\widetilde{Y}_h(s) = (M_X(s, h-1), M_X(s, h-2), \ldots, M_X(s, 0))$, where $M_X(s, i)$ maps $(s, i)$ to the past value of $X$ from $i$ steps before the defender reached the state $s$. |