# OpenReview forum: "General Markov Model for Solving Patrolling Games"
_auai.org/UAI/2024/Conference — UAI 2024 poster_

### Official Review · Reviewer_xLak · 2024-03-21

**Q2-1 Originality-Novelty:** 3
**Q2-2 Correctness-Technical Quality:** 3
**Q2-5 Clarity Of Writing:** 3

**Q10 Ethical Concerns:**

No.

**Q1 Summary And Contributions:**

This paper studied the stochastic patrolling problem in a model with both expressive power and computation friendliness. The paper proposed an algorithm solved in a linear problem for the stochastic patrolling problem and showed advantageous over the state-of-the-art algorithm.

**Q2-3 Extent To Which Claims Are Supported By Evidence:**

3: Good: the main claims are supported by convincing evidence (in the form of adequate experimental evaluation, proofs, (pseudo-)code, references, assumptions).

**Q2-4 Reproducibility:**

2: Fair: key resources (e.g. proofs, code, data) are unavailable but key details (e.g. proof sketches, experimental setup) are sufficiently well-described for an expert to confidently reproduce the main results.

**Q3 Main Strengths:**

The paper proposed a setup for stochastic patrolling problem with expressive power and computation tractability. The proposed algorithm for  optimal strategy can be solved in linear problem.

**Q4 Main Weakness:**

Experiments to show the advantage of the proposed algorithm are limited, only one comparison.

**Q5 Detailed Comments To The Authors:**

There is one typo on (1a) (max 0?) of the linear problem setup  on page 7.
There is one typo S or (S, A)? on line 2 of the first paragraph in section 4.2.

**Q9 Complying With Reviewing Instructions:**

Yes

---

> ### Author Rebuttal · Authors · 2024-04-08
>
> Thank you for the encouraging comments. We fixed the typos in the revised version of the manuscript that was published on an anonymous Google Drive
>
> https://drive.google.com/drive/folders/14pMy2Sq2b3cTWp2Y9eGgPCW_ebXXLqlD
>
> in a file patrolling_uai.pdf. The drive also contains animations of optimal surveillance strategies for a single and for two patrolling units for a scenario described in the running example. These illustrate the usefulness of a multi-agent setting and of a coverage function.
>
> W.r.t. max 0 in the line 1a (2a in the revised manuscript): it is not a typo, as in this version of the problem we only check the feasibility of the solution, i.e., we check whether there exist the stationary distribution and transition matrix satisfying all conditions.
>
> In fact, the absence of the objective function allows for an extension of the linear program to look for strategies with additional properties, i.e. with better mixing times. We plan to elaborate on that in a journal version of the paper.

---

### Official Review · Reviewer_SkKK · 2024-03-22

**Q2-1 Originality-Novelty:** 2
**Q2-2 Correctness-Technical Quality:** 2
**Q2-5 Clarity Of Writing:** 1

**Q1 Summary And Contributions:**

The paper deals with the problem of designing optimal patrolling strategies for mobile users. Protecting critical infrastructures is getting more and more importance in recent years and the advancing technology is allowing much more sophisticated attacks. Thus, any contribution towards an optimal use of limited security resources is of great relevance. Under realistic assumptions the problem can be described as a Bayesian game-theoretic problem in extensive form among two players, the defender and the attacker: the defender decides her patrolling strategy, the attacker observes the strategy of the defender and then she decides how and when to attack.

The main result of the paper is a general Markov model to describe the patrolling game. This model is based on a hidden Markov model with memory and combines ideas from stochastic patrolling, Stackelberg security games and Partially Observable Stochastic Games. Then a linear programming algorithm, called SHIELD, is presented to compute optimal defender strategies under the hypothesis that the strategy space admits a fixed hidden Markov model structure.

A few experiments are presented to validate the efficiency of the proposed algorithm.

**Q2-3 Extent To Which Claims Are Supported By Evidence:**

2: Fair: the main claims are somewhat supported by evidence (but the experimental evaluation may be weak, or does not match entirely with the claims, important baselines may be missing, proofs contain important ideas but lack rigor, algorithmic details are only discussed superficially, references are imprecise, assumptions are not sufficiently motivated or explicated, etc.).

**Q2-4 Reproducibility:**

1: Poor: key details (e.g. proof sketches, experimental setup) are incomplete/unclear, or key resources (e.g. proofs, code, data) are unavailable.

**Q3 Main Strengths:**

The paper studies a problem of great practical relevance.

**Q4 Main Weakness:**

The paper is very dense and hard to read. The presentation of the model is long and articulated and, even if a running example is presented to illustrate the different steps of the discussion, it is not sufficient to clarify all  the complexities of the model.

Experimental section is weak. Essentially, there is only one test on a single instance to compare the proposed algorithm with respect to one algorithm present in the literature.

**Q5 Detailed Comments To The Authors:**

What is the C_i in the definition of the game value?

It seems to me that you are considering the value of the game from the point of view of the defender. Are all the definitions in the paper consisting with this point of view? (for example, the game value against the actionable observation I?

Why are you restricting your attention to shift-invariant strategies for the defender? Is this without loss of generality?

Could you better clarify the last sentence of section 4.1? Why can you focus only on defender strategies with hidden Markov models?

**Q9 Complying With Reviewing Instructions:**

Yes

---

> ### Author Rebuttal · Authors · 2024-04-08
>
> In response to reviewers comments the paper was vastly improved. We published an updated version of the paper on an anonymous Google Drive
>
> https://drive.google.com/drive/folders/14pMy2Sq2b3cTWp2Y9eGgPCW_ebXXLqlD
>
> in a file patrolling_uai.pdf. We believe that the revised version is much easier to read.
>
> The drive also contains animations of optimal surveillance strategies for a single and for two patrolling units for a scenario described in the running example. These illustrate the usefulness of a multi-agent setting and of a coverage function.
>
> We rewrote proofs (included in the Appendix) with much more detail and added additional explanations through the text to make the paper more readable. In particular, there is a new Section 3.2 “The game dynamics” as well as a new detailed synthetic example in Section 5.1 (lines 510-551 and Figure 2).
>
> As for the experimental section, we compare ourselves directly to the work from the literature that is most closely related to ours (John et al. [2023]), as we can express the exact same experimental setting using our model.
>
> Another work with a setting relatively similar to ours is Horák et al. [2023], who perform simulations on random networks, just as we do. However, they employ a discount factor, hence we cannot compare our results directly.
>
> In general, the literature on patrolling games lacks a clearly defined set of benchmarks. One of the by-products of the presented research is such a set, along with a Monte Carlo method of evaluation of algorithm performance. This opens the door for applications of modern machine learning methods in security games and it is an exciting development in itself. We intend to publish such a benchmark along with the source code used in the experiments described in this paper.
>
> In the definition of the game value, C_i is the cone of path i defined in lines 144-145 in Section 3. (This refers to the updated version of the paper.) In other words, the game value considers all possible continuations of the path i and selects the most pessimistic for the defender (the use of inf).
>
> Indeed, we consider the value of the game from the point of view of the defender and, as mentioned above, we assume that the attacker makes the most damaging decision for the defender. In the security setting, it allows us to optimize against the worst possible scenario. The discussion of this aspect of our model can be found in Section 3.2 of the revised version of the manuscript.
>
> Regarding the restriction to shift-invariant strategies, at first we do assume it to be on the safe side, because a defender’s strategy that depends on time may be vulnerable to a properly timed attack, and in the definition of the game value against I (V_I(\mu)), the attacker picks a time of attack that is most favorable to him.
>
> But foremost, it is motivated by a theorem that is a subject of our ongoing research: we can prove that for each discrete strategy \mu (we say that a strategy is discrete if the range of the behavioral strategy is finite) and for each set of actionable observations I there exists a shift-invariant strategy \nu such that V_I(\nu) >= V_I(\mu). Please note that it is an open question whether every continuous strategy can be approximated by a discrete strategy with a close game value. Furthermore, the strategy \nu described above admits a hidden Markov model, which explains why we focus only on defender strategies of this type. We will publish this result in a journal version of the paper.

---

### Official Review · Reviewer_JkRR · 2024-03-22

**Q2-1 Originality-Novelty:** 3
**Q2-2 Correctness-Technical Quality:** 3
**Q2-5 Clarity Of Writing:** 3

**Q1 Summary And Contributions:**

In patrolling games, a defender and an attacker proceed along routs in a network. Each pair (p,j) of defender and attacker routes has a payoff, which intuitively indicates how well the attack j has been defended by p. In stochastic patrolling, the defender chooses its route randomly. A simple setting in the stochastic patrolling assumes that the strategy of the defender is given Markov decision process.
Another parameter to the setting concerns the observability of the defender by the attacker. Previous work use partially observable stochastic games (POSGs) in order to model the interaction between the attacker and the defender.

The paper introduces and studies a model that combines several methods, and suggests an algorithm that is based on this method and generates optimal strategies for the defender. It also contributes some general theoretical observations about such optimal strategies.

**Q2-3 Extent To Which Claims Are Supported By Evidence:**

3: Good: the main claims are supported by convincing evidence (in the form of adequate experimental evaluation, proofs, (pseudo-)code, references, assumptions).

**Q2-4 Reproducibility:**

3: Good: key resources (e.g. proofs, code, data) are available and key details (e.g. proofs, experimental setup) are sufficiently well-described for competent researchers to confidently reproduce the main results.

**Q3 Main Strengths:**

The problem is well motivated, and the suggested method includes nice ideas and performs well in the experiments.

**Q4 Main Weakness:**

The paper is of an incremental nature.
The paper is very well written in the high level (in particular, the running example is very helpful), but the technical details are very difficult to follow.

**Q5 Detailed Comments To The Authors:**

The paper is very well written in the high level (in particular, the running example is very helpful), but the technical details are very difficult to follow.

**Q9 Complying With Reviewing Instructions:**

Yes

---

> ### Author Rebuttal · Authors · 2024-04-08
>
> We agree that some of the technical details might be hard to follow, and we thank the reviewer for appreciating the running example. We hope that implementing the comments of the reviewers in the revised version of the manuscript makes the paper more comprehensible to the reader. It was published on an anonymous Google Drive
>
> https://drive.google.com/drive/folders/14pMy2Sq2b3cTWp2Y9eGgPCW_ebXXLqlD
>
> in a file patrolling_uai.pdf. The drive also contains animations of optimal surveillance strategies for a single and for two patrolling units for a scenario described in the running example. These illustrate the usefulness of a multi-agent setting and of a coverage function.
>
> We rewrote proofs (included in the Appendix) with much more detail and added additional explanations through the text to make the paper more readable. In particular, there is a new Section 3.2 “The game dynamics” as well as a new detailed synthetic example in Section 5.1 (lines 510-551 and Figure 2).

---

### Official Review · Reviewer_6929 · 2024-03-23

**Q2-1 Originality-Novelty:** 3
**Q2-2 Correctness-Technical Quality:** 3
**Q2-5 Clarity Of Writing:** 2

**Q1 Summary And Contributions:**

The paper studies a Markov model for the patrolling security game that generalizes the setup in the existing literature. Specifically, the authors provide a framework where both the defenders and the attackers can have a history dependent strategy and thus make the considered game model richer. Under certain assumptions, approximation algorithms as well as an upper bound on the game value are obtained.

**Q2-3 Extent To Which Claims Are Supported By Evidence:**

3: Good: the main claims are supported by convincing evidence (in the form of adequate experimental evaluation, proofs, (pseudo-)code, references, assumptions).

**Q2-4 Reproducibility:**

3: Good: key resources (e.g. proofs, code, data) are available and key details (e.g. proofs, experimental setup) are sufficiently well-described for competent researchers to confidently reproduce the main results.

**Q3 Main Strengths:**

1. The authors provide a more general yet still tractable formulation of the patrolling security game.
2. Strategy synthesis algorithm was given accompanied with rigorous theoretical analysis.

**Q4 Main Weakness:**

Some parts of the paper is hard to follow and require more explanation.

**Q5 Detailed Comments To The Authors:**

1. More explanations may be given to "We equip (S,A) with a graph homomorphism X:S\to L, called projection, which translates defender’s internal scheduling to actions in the physical space (L,R)." Since the projection is from the internal strategy state to a physical location, What does it mean by "translates defender’s internal scheduling to actions"? Later in Section 4.4, this projection is defined to be "let X be a projection from (S,A) to (L,R) defined in Section 4.1".

2. "Let S be a strategy state and action space", isn't S just the strategy state?

3. "Let Z be an arbitrary function on S." What is the image of Z?

4. "Thus, the current internal state s\in S of the defender determines uniquely what attacker’s observation was t steps ago." How does the internal strategy state relate to the observations of the attackers?

5. What is "rg(Z)"? Section 4.2 is a bit difficult to understand. What does the memory mean exactly?

6. The shift operator may be more explicitly defined.

7. V is used to denote the game value as well as the value of targets.

8. It is not clear what "Let S be a strategy state and action space" means.

9. What are X and Y in "strategy space (S,A) has memory of length at least \tau_j with respect to X and Y"?

10. What is the meaning of "Notice that the payoff function G_j of the attack plan j depends only on the last \tau_j states of the physical space"? The payoff function was defined to be a mapping from \matchal{L}\times T to \mathbb{R}. In the next sentence, G_j is a function of a finite-length sequence of physical locations.

11. What is M_Y?

12. Aside from all the technical descriotions, is it true that the game model is essentially equivalent to a Stacklberg game model where the defender uses a Markov chain with memory (a higher order Markov chain) and the attacker uses an attack strategy that depends on the history of the defender? If not, what are the key differences?

13. It seems that G_j^{i-1} should be G_j^{i}. For example, when the attacker attacks after obtaining a single observation, i.e., i=1, the payoff for the defender should be G_j^{1} instead of G_j^{0}? Note that it was assumed that attack begins at the moment when observation i ends.

14. In the defition of V(\mu), the infimum should be taken over L^* \cup \emptyset to accomodate the case of no observation? Note that L^* is defined to be a set of finite nonempty paths. In fact, in the defition of $P(i~p)$ for i,p\in L^*, it seems that i=\epsilon (empty string?) is allowed.

15. C_i is not defined in the definition of the game value.

16. In 2.1.2, "there exists \tau_j \in N such that" is not clear. Is this \tau_j the same as the \tau_j in the definition of the attack plan? Is \tau_j a fixed value of a decision variable for the attacker?

17. T was the attack plan in the modeling section, but the targets in the patrolling game.

18. The definition of shift-invariance is not clear. A crucial condition in [Sarig, 2009, Proposition 1.8] is missing in the paper. Simply saying \mu(L\dot A) = \mu(A) may cause confusion.

18. It is a bit misleading to call (L,R) a physical space in Section 3.2 since it is actually a tensor product of "physical spaces" of multiple agents.

19. In "Then, the physical space (L,R) is defined to be a tensor product of subdivided graphs", what are subdivided graphs?

20. What is x in D_j(x) in Section 3.2?

21. In section C.1, "Consider the set of attack plans T to be a subset of L", since L is a product space, the attack plan cannot be a subset of L.

13. typos: "a the same time"; "with locations (L,R) representing"; "scenario An"; "assumption it the setting"; "the duration the attack plan"; "has has a hidden"; "AN INSTANCE OUR GENERAL MODEL"; "unit until"

**Q9 Complying With Reviewing Instructions:**

Yes

---

> ### Author Rebuttal · Authors · 2024-04-08
>
> We are grateful for a comprehensive review. We believe that we fixed all of the issues that were raised. Because there were so many, we published an updated version of the paper on an anonymous Google Drive
>
> https://drive.google.com/drive/folders/14pMy2Sq2b3cTWp2Y9eGgPCW_ebXXLqlD
>
> in a file patrolling_uai.pdf. The file contains line numbers that we reference in a detailed response below. All line and Section numbers below refer to the updated version of the paper.
>
> We rewrote proofs (included in the Appendix) with much more detail and added additional explanations through the text to make the paper more readable. In particular, there is a new Section 3.2 “The game dynamics” as well as a new detailed synthetic example in Section 5.1 (lines 510-551 and Figure 2).
>
> The original idea of the paper was to compute defender strategies using Markov chains with memory. Then we found out that what really is needed is an abstracted property that we call memory of a state and action space. We consider this a breakthrough: this property is held by many more spaces than Markov chains with memory, as we describe in Section 5.3. Markov chains are usually applied in robotic surveillance; there is another paradigm of Bayesian Stackelberg games that is applied to security; this can be modeled in our approach using a strategy space with a large girth. In a way, our approach unifies both paradigms.
>
> Note that the proof of correctness of the algorithm is long and tedious, but at the end we did independent verification of correctness of game values computed with SHIELD using Monte Carlo simulations. It is interesting to note that if the hidden Markov model is irreducible, then the constructed measure is ergodic and the Monte Carlo simulation is very effective.
>
> We construct shift-invariant strategies to be on the safe side, as argued in lines 358-369. However, as mentioned in lines 374-382, we can prove that any discrete strategy (see line 377) can be modified to a shift-invariant strategy without a loss in the game value. It is a highly non-trivial result. We do not know if every continuous strategy can be approximated by a discrete one (at a small loss of game value).
>
> All of the above is the subject of ongoing research and we intend to elaborate on it in a journal version of the paper.
>
> We uploaded to Google Drive two animations to illustrate the usefulness of both multi-agent setting and of a coverage function.
>
> Detailed answers
>
> 1. Homomorphism X translates change in the hidden state into movements of patrolling units. Please see an example in lines 510-551.
> 2. Fixed (line 504).
> 3. We do not put any a priori constraints on the codomain of Z (line 575).
> 4. We assume that the attacker bases his decision to attack on an observation of recent movements of patrolling units; what he wishes to observe is a property of the attacker type and is specified in a set of actionable observations I. See discussion in Section 3.3.1 and lines 278-289.
> 5. rg(Z) is the range of Z (line 575). Memory of length t is a property of the strategy space: a state s in S uniquely determines what the value of Z was t-1 units of time ago. See lines 581-588 and Lemma 5.2.
> 6. Fixed (lines 122-125 and 174-177).
> 7. Game value changed to \mathcal{V}.
> 8. Fixed (lines 500-509).
> 9. X is the projection from S to L and Y is the attacker's observation function (line 643-645).
> 10. An assumption that attack resolves in a finite number of steps allows us to compute G_j from a finite prefix of an infinite patrol schedule (lines 338-346).
> 11. M_Y is the memory function of Y, defined in Lemma 5.2.
> 12. Please see a comment before the “Detailed answers” section.
> 13. The attack begins at the turn when the observation ends, e.g. if the attacker decides to attack a target that is protected he is caught immediately.
> 14. We clarified that and introduced notation L^*_+ to denote a set of non-empty paths (lines 181-183). The attacker cannot attack at history \epsilon because at \epsilon the location of patrolling units is not defined.
> 15. C_i is a cone of i, see lines 144-145.
> 16. \tau_j is the time required to successfully penetrate target j and it is part of the model of the critical infrastructure. \tau_j in Section 3.3.3 and \tau in Section 4.2 are the same.
> 17. We think that these are interchangeable: each attack plan corresponds to an attacked target. (line 410).
> 18. We define shift-invariance as in [Sarig, Def. 1.2.2 (p. 12)]. We use [Sarig, Proposition 1.8] to argue that a Markov measure ([Sarig, Definition 1.8]) defined using transition matrix N and a stationary distribution \sigma is shift-invariant (lines 552-559). We use \sigma to select the first state in a defense plan.
> 19. We clarified this in lines 131-133: “Note that we will commonly use a single element of S to represent a location of multiple patrolling units”.
> 20. See Section A.1.
> 21. Fixed (we meant D_j(p)), lines 491 and 499.
> 22. Fixed.

---

### Meta-Review · Area_Chair_mcyu · 2024-04-15

Four reviews have been obtained. Two reviewers recommend acceptance, one reviewer recommends weak acceptance, and the other one recommends borderline acceptance. Overall, all reviewers are positive about the paper, although they raised several main weaknesses regarding the technical presentation and limited numerical experiments. The average confidence of the reviewers is not high; so after a second look over the paper, I believe the paper is of good technical quality and presents a novel approach for modeling and solving patrolling games.